# Loss of the transcription factor RBPJ induces disease-promoting properties in brain pericytes

Rodrigo Diéguez-Hurtado [1,2], Katsuhiro Kato[1], Benedetto Daniele Giaimo[3], Melina Nieminen-Kelhä[4,5], Hendrik Arf [1], Francesca Ferrante[3], Marek Bartkuhn[6], Tobias Zimmermann[7], M. Gabriele Bixel[1], Hanna M. Eilken[1,8], Susanne Adams[1], Tilman Borggrefe[3], Peter Vajkoczy[4,5] & Ralf H. Adams [1,2]

Sufficient vascular supply is indispensable for brain development and function, whereas dysfunctional blood vessels are associated with human diseases such as vascular malformations, stroke or neurodegeneration. Pericytes are capillary-associated mesenchymal cells that limit vascular permeability and protect the brain by preserving blood-brain barrier integrity. Loss of pericytes has been linked to neurodegenerative changes in genetically modified mice. Here, we report that postnatal inactivation of the *Rbpj* gene, encoding the transcription factor RBPJ, leads to alteration of cell identity markers in brain pericytes, increases local TGFβ signalling, and triggers profound changes in endothelial behaviour. These changes, which are not mimicked by pericyte ablation, imperil vascular stability and induce the acquisition of pathological landmarks associated with cerebral cavernous malformations. In adult mice, loss of *Rbpj* results in bigger stroke lesions upon ischemic insult. We propose that brain pericytes can acquire deleterious properties that actively enhance vascular lesion formation and promote pathogenic processes.

[1] Department of Tissue Morphogenesis, Max Planck Institute for Molecular Biomedicine, Röntgenstrasse 20, 48149 Münster, Germany. [2] Faculty of Medicine, University of Münster, 48149 Münster, Germany. [3] Institute of Biochemistry, University of Giessen, Friedrichstrasse 24, 35392 Giessen, Germany. [4] Department of Neurosurgery, Charité-Universitätsmedizin, Charitéplatz 1, Berlin 10117, Germany. [5] Berlin Institute of Health, Charitéplatz 1, 10117 Berlin, Germany. [6] Institute for Genetics, University of Giessen, Heinrich-Buff-Ring 58-62, 35392 Giessen, Germany. [7] Bioinformatics and Systems Biology, University of Giessen, Heinrich-Buff-Ring 58-62, 35392 Giessen, Germany. [8]Present address: Bayer AG, Aprather Weg 18a, 42113 Wuppertal, Germany. Correspondence and requests for materials should be addressed to P.V. (email: peter.vajkoczy@charite.de) or to R.H.A. (email: ralf.adams@mpi-muenster.mpg.de)

T he brain vasculature has unique characteristics strongly related to the metabolic and physiologic properties of the surrounding tissue. As the brain consumes large amounts of energy on demand without the possibility of energy storage, regional and dynamic differences in brain activity are matched by changes in blood flow[1]. At the same time, exchange between the circulation and brain tissue is tightly controlled by the blood–brain barrier (BBB), which prevents neuronal damage and is critical for brain homeostasis[2]. Pericytes are capillary-associated vessel wall (mural) cells and, together with neural cells and endothelial cells (ECs), part of the so-called neurovascular unit (NVU), the structure underlying the BBB[3]. Pericyte deficiency in mice increases BBB permeability, which involves changes in EC gene expression and enhanced transport of vesicles across the EC monolayer (i.e., endothelial transcytosis) without compromising endothelial cell–cell junctions[4]. One of the changes in brain ECs that results from the loss of pericytes is the downregulation of *Mfsd2a*, which encodes a transmembrane transport protein. Interestingly, endothelial transcytosis is increased in *Mfsd2a*-deficient mice leading to BBB defects[5]. Pericyte contractility has been implicated in the regulation of cerebral blood flow[6], whereas other data attributes this function to arteriolar smooth muscle cells (SMCs)[7]. Pericyte loss in mutant mice has been also associated with a number of neuropathologies such as white matter disease, neurovascular uncoupling, Alzheimer's disease, and age-dependent memory impairment[8–10].

Here, we report that loss of the transcription factor RBPJ alters fundamental aspects of brain pericyte identity, which involves increased contractility, capillary obstruction and the formation of aneurysms, increased TGFβ activation, haemorrhaging, and NVU dysfunction. Our data argue that these processes are triggered by the loss of RBPJ repressor activity and not because of its important and well-known role as a transcriptional activator in the Notch signalling pathway[11]. Strikingly, *Rbpj* deficiency in pericytes induces the acquisition of vascular lesions resembling cerebral cavernous malformations (CCMs) and has other detrimental effects, which are not recapitulated by mice lacking pericytes. We therefore propose that pericytes can acquire disease-promoting properties, which lead to vascular malformations in the brain and increased tissue damage after ischaemic injury.

## Results

**Inducible gene targeting in mouse brain pericytes**. In order to genetically target pericytes in vivo, we have recently developed *Pdgfrb-CreERT2* transgenic mice, which were shown to work efficiently in mural cells of the postnatal retina[12], an extracranial part of the central nervous system (CNS). Analysis of the cerebral vasculature from these mice in combination with the *Rosa26-mTmG* Cre reporter allele[13] showed efficient and mural cell-specific recombination (i.e., GFP expression) throughout the brain after tamoxifen administration during embryonic or early postnatal development (Supplementary Fig. 1a, b). Recombination efficiency in cortical regions of the cerebrum of young pups (P10) was around 80% (Supplementary Fig. 1c) and no obvious differences were found among distinct regions of the brain (Supplementary Fig. 1d). Likewise, efficient targeting of mural cells was achieved after tamoxifen induction in juvenile and adult animals (Supplementary Fig. 1e).

**Rbpj-deficient pericytes impair brain vascular morphogenesis**. To address the role of RBPJ in mural cells, we generated tissue-specific *Rbpj* conditional knockouts (*Rbpj*iPC; Fig. 1a) by interbreeding of *Pdgfrb-CreERT2* and *Rbpj*lox/lox animals[14]. Inducible inactivation of *Rbpj* mediated by *Pdgfrb-CreERT2*-driven

recombination resulted in widespread focal haemorrhaging in the brain with higher prevalence in cortical regions of the cerebrum and cerebellum, and less prevalent or completely absent in the hippocampal formation, brain stem (thalamus and hypothalamus), and olfactory bulb (Fig. 1b–d and Supplementary Fig. 1f, g). Notably, despite early and efficient deletion of *Rbpj* with tamoxifen injection from postnatal day 1 (P1) to P3, the vascular lesions were restricted to the CNS (Supplementary Fig. 1h), started to develop at P7, and were prominent by P10. Moreover, animals with hemizygous deletion of *Rbpj* showed no phenotypic alteration, were undistinguishable from Cre-negative littermates (Supplementary Fig. 2a–h), and were therefore used as controls in experiments that required Cre-induced expression of reporter alleles.

Microscopic and quantitative analysis of the brain cortical vasculature revealed an increase in vascular area together with a reduction of vascular density in *Rbpj*iPC animals (Fig. 1e). This was due to a significant enlargement of veins and capillaries together with a decline in the number of side branches, whereas arteries remained mostly unaffected in terms of size and morphology (Fig. 1f, g). In particular, veins, venules, and the surrounding capillaries were highly tortuous and showed local constrictions leading to the formation of aneurysms in adjacent vessel segments (Supplementary Fig. 3a). Furthermore, analysis of the vasculature in the pial surface of the brain cortex by systemic perfusion of coloured dyes revealed the existence of arteriovenous malformations, which bypass capillaries and allow direct shunting of arterial blood into veins (Fig. 1h and Supplementary Fig. 3b).

It has been previously reported that the vascularization of the CNS occurs mainly via sprouting angiogenesis, which is initiated in the embryo and continues during postnatal stages in order to expand and remodel the brain vasculature[15]. Mural cell-specific *Rbpj* mutant animals showed a remarkable reduction in the number of sprouts and instead of the usual tip-cell morphology, characterized by extension of long filopodia, emerging capillaries were blunt-ended and often dilated resembling microaneurysms (Fig. 1i, j). In addition, the total number and density of EC nuclei, identified by expression of the transcription factor ERG, was strongly increased both in capillaries and veins from early stages (P7) onward (Fig. 1k and Supplementary Fig. 3c, d). EdU administration revealed a >2-fold increase in EC proliferation (Fig. 1l and Supplementary Fig. 3e), leading to the abnormal accumulation of ECs with superimposed nuclei within the twisted and tangled *Rbpj*iPC capillaries (Fig. 1m). At the molecular level, RT-qPCR gene expression analysis of freshly sorted ECs from brain cortex revealed a strong downregulation in the expression of *Dll4* and *Esm1*, known markers of sprouting endothelial tip cells[16], and a significant upregulation of *Myc*, a powerful driver of EC proliferation[17] (Fig. 1n). Transmission electron microscopy confirmed the strong EC hyperplasia together with the emission of intraluminal protrusions, enlargement of the sub-endothelial basement membrane, and formation of intra-cytoplasmic canaliculi in the *Rbpj*iPC endothelium (Fig. 2a, b). In contrast, the ultrastructure of EC junctions was maintained in line with normal expression of tight junction and adherens junction proteins (Fig. 2a and Supplementary Fig. 3f, g). Likewise, vascular ensheathment by astrocyte endfeet was maintained even in areas with haemorrhages (Fig. 2c, d). Noteworthy, degenerative changes in the endothelium of *Rbpj*iPC mice, such as accumulation of pinocytotic vesicles and cytoplasmic vacuolization, were obvious already at P7 and were substantially worse by P10 (Supplementary Fig. 3h). These changes may compromise EC integrity and can be related to the increased EC apoptosis detected at P10 (Supplementary Fig. 3i, j)

The changes in blood vessel organization and, in particular, the enlargement of capillaries and veins described above induce a

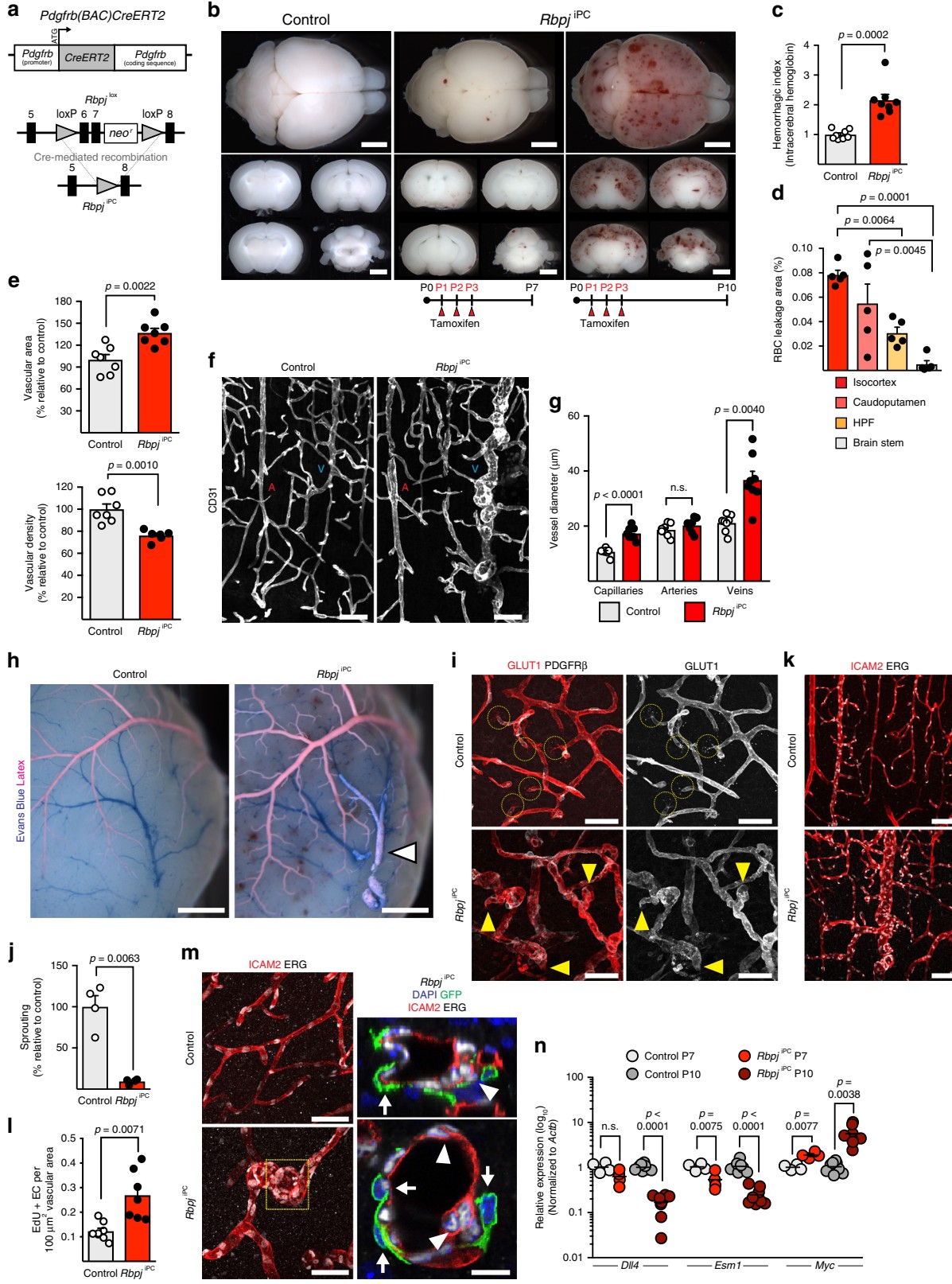

strong reduction in blood flow velocity in these vessels, as revealed by two-photon in vivo imaging of the brain vasculature, without strong alterations in the arterial side of the network (Fig. 2e, f).

Remarkably, the *Rbpj*iPC vascular anomalies were not related to changes in pericyte coverage, which remained unaltered (Fig. 2g,

h), nor to differences in recombination efficiency between homozygous mutants and heterozygous controls (Supplementary Fig. 3k). In addition, *Pdgfrb-CreERT2*-controlled acute ablation of mural cells by induced expression of diphtheria toxin in *ROSA-DTA* knock-in mice[18] failed to cause similar phenotypic outcomes despite high efficiency of mural cell depletion

**Fig. 1** *Pdgfrb-CreERT2*-mediated *Rbpj* deletion compromises brain vessels. **a** *Pdgfrb-CreERT2* transgene and Cre-mediated recombination of *Rbpj*[lox] allele. **b** Whole brains and coronal sections from control and *Rbpj*[iPC] mice at P7 or P10 after tamoxifen administration (P1-P3). Scale bar, 2 mm. **c** Haemorrhagic index. *p*-values, Mann–Whitney *U*-test, *n* = 8. **d** Red blood cell (RBC) extravasation in brain parenchyma. HPF, hippocampal formation. *p*-values, one-way ANOVA and Tukey's test, *n* = 5. **e** Vascular area and density in brain cortices. *p*-values, Student's *t*-test, *n* = 6–7. **f** Confocal images of blood vessels (CD31[+], white) in brain cortices. Arteries (A) and veins (V) are indicated. Scale bar, 100 μm. **g** Quantitation of vessel diameter in cortical blood vessels. *p*-values, Brown–Forsythe and Welch one-way ANOVA with Tamhane's T2 test, *n* = 4. n.s., not statistically significant. **h** Arteriovenous malformation in brain superficial vasculature as revealed by the presence of latex in the caudal rhinal vein (arrowhead). Scale bar, 1 mm. **i** Confocal images of cortical blood vessels stained for GLUT1 (red, white) and PDGFRβ (white). Normal sprouts (dashed circles) replaced by blunt-ended capillaries (yellow arrowheads) in *Rbpj*[iPC] mice. Scale bar, 50 μm. **j** Quantitation of sprouting in brain cortices. *p*-values, Welch's *t*-test, *n* = 4. **k** Confocal images of blood vessels (ICAM2[+], red) and EC nuclei (ERG[+], white) in brain cortices. Scale bar, 100 μm. **l** Quantitation of EC proliferation in blood vessels of P7 mice. *p*-values, Welch's *t*-test, *n* = 7–8. **m** Confocal images of cortex capillaries stained for ICAM2 (red), ERG (white), and GFP (green, recombined mural cells). Right panel shows single optical sections of boxed inset in *Rbpj*[iPC] (left column). Recombined pericytes (white arrows) around tangled capillaries with superimposed EC nuclei (white arrowheads). Scale bar, 50 μm (left panels) and 10 μm (high magnifications). **n** RT-qPCR analysis in sorted ECs from brain cortices of P7 and P10 mice. *p*-values, Brown–Forsythe and Welch one-way ANOVA with Tamhane's T2 test for *Dll4* and *Myc*; one-way ANOVA with Sidak's test for *Esm1*, *n* = 4–8. All images correspond to P10 mice unless otherwise indicated. Error bars represent s.e.m. Source data are provided as a Source Data file

(Supplementary Fig. 4a–e). Likewise, chronic paucity of pericyte coverage in mice lacking the retention motif of the growth factor PDGF-B (*Pdgfb*[Ret/Ret] mutants[19]) did not phenocopy the *Rbpj*[iPC] vascular architecture, endothelial sprouting defects, and haemorrhaging (Supplementary Fig. 4f–j).

Secondary to the *Rbpj*[iPC] vascular defects, noticeable changes in other cellular components of the NVU were observed. Reactive astrogliosis, a typical response to CNS injury, was detected by upregulated immunostaining of glial fibrillary acidic protein (GFAP) (Supplementary Fig. 5a, b). Infiltration and activation of Iba1[+] (ionized calcium binding adaptor molecule 1) microglia was strongly increased in *Rbpj*[iPC] brains (Supplementary Fig. 5c, d), and oedema, a common finding after focal cerebral ischaemia[20], was evident in the *Rbpj*[iPC] cortex and found in association with collapsed capillaries (Supplementary Fig. 5e). Neurons are especially sensitive to deprivation of oxygen or glucose and therefore to damages in vascular integrity[21]. Immunodetection of microtubule-associated protein 2 (MAP2), a dendritic marker, showed signal-depleted foci around severely affected *Rbpj*[iPC] blood vessels (Supplementary Fig. 5f). Likewise, NeuN and neurofilament-H (NF-H) immunostaining, markers for postmitotic neurons and axons, showed similar irregularities around *Rbpj*[iPC] vascular lesions (Supplementary Fig. 5g).

Together, these results show that *Rbpj* inactivation in mural cells induces severe vascular abnormalities that compromise NVU integrity without affecting pericyte coverage and through mechanisms that are not recapitulated by chronic or acute pericyte ablation.

**Rbpj maintains the molecular identity of brain pericytes.** To gain insight into the molecular alterations elicited by the inactivation of *Rbpj* in brain mural cells, we interbred the conditional *Rbpj*[iPC] mutant with *Rpl22*[tm1.1Psam] knock-in animals[22], which enable Cre-controlled hemagglutinin-tagging of the ribosomal protein Rpl22 (RiboTag) and thereby immunoprecipitation of actively translating, polyribosome-bound transcripts. Following tamoxifen administration from P1–P3, RNA from P7 and P10 control and *Rbpj*[iPC] brain cortices was isolated and sequenced in triplicates. Unsupervised hierarchical clustering and principal component analysis (PCA) of all RNA-seq datasets (Supplementary Fig. 6a, b) showed high reproducibility of the gene expression profiles among samples of the same group and a bigger difference between control and *Rbpj*[iPC] animals as the phenotype worsens at P10. Differential gene expression analysis with a false discovery rate (FDR)-adjusted *p*-value < 0.05 and an absolute log$_2$ fold change > 0.5 identified 450 differentially expressed genes (DEGs) at P7, of which 234 were upregulated and

216 downregulated, and 2551 DEGs at P10, 1402 of which were upregulated and 1149 downregulated (Fig. 3a, b and Supplementary Data 1 and 2). Given the drastic increase in DEGs at P10, we reasoned that secondary effects elicited by bleeding and inflammation could be important contributors at this stage and we therefore first focused on the incipient and more discrete changes detected at P7 for gene ontology (GO) analysis. GO analysis for biological processes linked the gene expression changes in P7 *Rbpj*[iPC] PDGFRβ[+] mural cells to vascular development, blood vessel morphogenesis and angiogenesis, while the extracellular matrix (ECM) and cell surface were the overrepresented cellular components (Fig. 3c). Next, we classified groups of genes with similar expression profiles across the different genotypes and stages analyzed. To this end, we applied model-based hybrid-hierarchical clustering[23] to 14,742 transcripts and focused on clusters that group together genes which are consistently up- or downregulated both at P7 and P10 (Fig. 3d). Interestingly, the group of downregulated genes included several commonly used pericyte markers, including *Pdgfrb*, *Anpep*, *Rgs5*, *Cspg4*, and *Notch3*, which were also reduced in freshly sorted *Rbpj*[iPC] brain mural cells analyzed by RT-qPCR (Fig. 3e and Supplementary Fig. 6c). Moreover, 30 out of 49 recently proposed pericyte markers based on single-cell RNA-seq[24] were downregulated in *Rbpj*[iPC] pericytes (Fig. 3f). At the same time, the cluster of consistently upregulated genes included numerous markers such as *Acta2* (αSMA) and *Tagln* (SM22α), which were recently shown to be at least 10-fold enriched in arterial/arteriolar SMCs relative to pericytes[24] (Fig. 3g, h).

Given that the *Pdgfrb-CreERT2* transgene triggers recombination in all mural cell types and to rule out that the increase in SMC markers might reflect a larger abundance of this mural cell type, we analyzed vessel-associated cells around arteries and arterioles. Remarkably, the coverage of vascular SMCs was reduced and irregular in *Rbpj*[iPC] brains (Fig. 4a, b), whereas mutant pericytes showed increased immunostaining for the contractility-related cytoskeletal proteins αSMA, SM22α, Vimentin, Desmin, and Nestin (Fig. 4c–g and Supplementary Fig. 3g). Moreover, phosphorylation of myosin light chain 2, which reflects the contractile activity of actomyosin, was strongly increased in *Rbpj*-deficient pericytes, whereas it was almost undetectable in control brain microvasculature (Fig. 4h). Increased contractility of pericytes was also seen in a collagen gel contraction assay and, likewise, the overexpression of *Acta2* and *Tagln* was confirmed in vitro after *Rbpj* inactivation in cultured pericytes (either by lentivirus-mediated Cre expression or treatment of cells with TAT-Cre protein) (Fig. 4i–k and Supplementary Fig. 6d). Consistent with the aforementioned changes in the expression

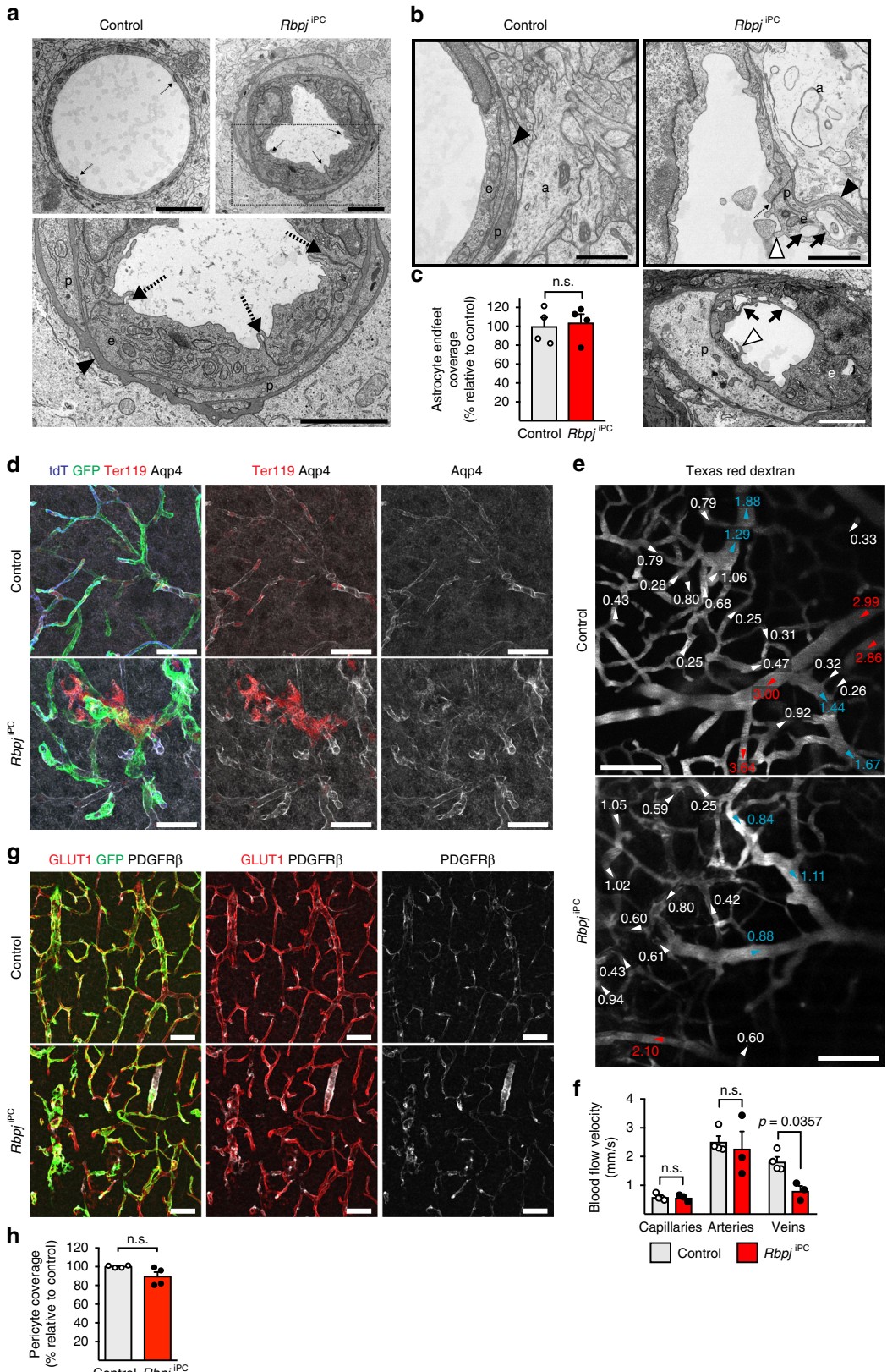

of cytoskeletal proteins, we noticed that mutant pericytes showed profound morphological changes with a marked increase in cellular projections and membrane protrusions (Fig. 4l).

Taken together, these data argue that the loss of *Rbpj* compromises pericyte identity, leading to increased contractility and the upregulation of vascular SMC markers.

**Notch signalling reduction fails to phenocopy *Rbpj* deletion.** Given the notorious differences in the phenotype elicited by *Rbpj* deletion in pericytes with respect to artery-associated SMCs, we next investigated the activation level of canonical Notch signalling by analyzing two different reporter mouse lines (*CBF:H2B-Venus* and *Hey1-EGFP*) in which fluorescent protein expression is

**Fig. 2** Ultrastructural changes and blood flow assessment in *Rbpj*iPC brains. **a** Electron micrographs of P10 brain cortex capillaries. Note EC hyperplasia and lumen deformation in mutant. EC junctions are not compromised (arrows). Bottom panel shows higher magnification of the boxed inset in *Rbpj*iPC for better appreciation of electron-dense and continuous junctions. The basement membrane between *Rbpj*iPC ECs (E) and pericytes (P) is notably enlarged (arrowhead). Scale bar, 2 μm. **b** Electron micrographs of mouse brain cortex capillaries at P10. ECs (E), pericytes (P), astrocyte endfeet (A), and basement membrane (black arrowhead) are indicated. Note vacuolization (black arrows) in *Rbpj*iPC samples as well as emission of luminal projection (white arrowheads), whereas EC junctions appear intact (dashed arrow). Scale bar, 1 μm (top panels) and 2 μm (bottom panel). **c** Quantitation of astrocyte endfeet coverage in P10 cortical blood vessels. *p*-values, Student's *t*-test, $n = 4$. n.s., not statistically significant. **d** Confocal images of brain cortex blood vessels (tdT+, blue) showing areas of red blood cells (Ter119+, red) extravasation which are not associated to defects in astrocyte endfeet polarization (Aqp4+, white) or absence of recombined mural cells (GFP+, green). Scale bar, 50 μm. **e** In vivo two-photon imaging of brain blood vessels from P10 mice visualized by intravenous injection of Texas Red-dextran (white). Arrowheads (white, red, or cyan for capillaries, arteries, or veins, respectively) indicate the direction of blood flow, velocity (mm s$^{-1}$) is annotated for specific vessel segments. Scale bar, 100 μm. **f** Quantitation of blood flow velocities of different brain vascular segments in P10 mice. *p*-values, one-way ANOVA with Sidak's test, $n = 3$–4. n.s., not statistically significant. **g** Confocal images of brain cortex blood vessels stained for GLUT1 (red), PDGFRβ (white), and GFP (green) in P10 mice. Note that despite of vessel defects in the knockout animals, mural cell coverage is not affected. Scale bar, 100 μm. **h** Quantitation of mural cell coverage in cortical vasculature at P10. *p*-values, Welch's *t*-test, $n = 4$. n.s., not statistically significant. Error bars represent s.e.m. Source data are provided as a Source Data file

controlled by RBPJ binding[25] or by expression of the Notch downstream gene *Hey1* (http://www.gensat.org/), respectively. Interestingly, both lines fail to show evidence for Notch signalling in brain pericytes during early stages of postnatal development (from P1 to P15) (Supplementary Fig. 7a, b), whereas SMCs covering penetrating arterioles in the cortex or pial arteries in the brain surface show reporter expression (Supplementary Fig. 7c, d). These results suggest that the phenotypic changes seen in *Rbpj*iPC pericytes are probably unrelated to the lack of Notch activation. In order to further characterize the relevance of canonical Notch signalling in pericytes, we studied the role of Notch family receptors in mural cells. Our RiboTag RNA-seq analysis had shown that *Notch3*, *Notch2*, and *Notch1*, in this order, are the most abundant receptors expressed in brain PDGFRβ+ cells during early postnatal stages (Supplementary Fig. 8a). *Notch3* constitutive and ubiquitous knockout mice[26] are viable and, despite of arterial SMC degeneration and focal disruptions in the BBB, pericytes remain unaffected[27]. Notch1 is not very active in mural cells, as reflected by very low frequency of recombination in mouse brain pericytes of the *Notch1*tm3(cre)Rko/J Cre reporter[28] in the *Rosa26-mTmG* background (Supplementary Fig. 8b). *Pdgfrb-CreERT2*-mediated inactivation of *Notch2* in a global *Notch3*−/− background[26] had also no effect on pericytes and brain vascular morphogenesis (Supplementary Fig. 8c–e). Furthermore, *Pdgfrb-CreERT2* controlled expression of dominant negative Mastermind-like 1 in *Rosa26*dnMaml1 mice[29] (hereafter, *Maml1*dnPC), which blocks all Notch-mediated activation of the RBPJ-associated transcriptional activator complex, impaired expression of the Notch target gene *Hey1* in freshly sorted brain mural cells but failed to induce obvious changes in vascular organization, pericyte abundance, EC density, or endothelial sprouting (Supplementary Fig. 8f–l). The expression of tip cell markers and *Myc* was also not significantly altered in sorted *Maml1*dnPC brain ECs. Moreover, the expression of known pericyte markers, strongly compromised in *Rbpj*iPC mutants, was not significantly changed in *Maml1*dnPC brains (Supplementary Fig. 8m, n). Finally, we tested the effects of *Pdgfrb-CreERT2*-mediated expression of constitutively active Notch1 intracellular domain in *Gt(ROSA)26Sor*tm1(Notch1)Dam mice[30]. The resulting *NICD*iPC gain-of-function mutants showed no appreciable alteration in the expression of the mural cell markers PDGFRβ, SM22α, Desmin, and CD13. Vascular area, pericyte coverage, or the frequency of endothelial sprouting were comparable in *NICD*iPC and control littermates (Supplementary Fig. 9a–g).

These data establish that alterations in canonical Notch signalling do not impair proper vascular morphogenesis in the postnatal brain, indicating that the deleterious effects of *Rbpj* inactivation in mural cells arise independently from Notch.

**Rbpj-deletion alter ECM composition and TGFβ signalling**. In addition to its role in Notch signalling, RBPJ can also repress gene expression[31]. As Notch pathway mutants or mural cell ablation failed to reproduce the *Rbpj*iPC phenotype, we hypothesized that the functional changes in RBPJ-deficient pericytes are caused by the upregulation of transcripts that are normally expressed at low level. Gene set enrichment analysis (GSEA) of consistently upregulated genes in the P7 and P10 RiboTag RNA-seq data revealed significant upregulation of TGFβ signalling targets in *Rbpj*iPC PDGFRβ+ cells compared to controls (Fig. 5a). Immunostaining of *Rbpj*iPC brain sections confirmed a striking increase in phosphorylated SMAD3 (Fig. 5b, c), the intracellular signal transducer and transcriptional modulator activated by TGFβ, relative to control. Increased phosphorylation of SMAD1/5 was also detected (Fig. 5d, e) indicating elevated signalling through bone morphogenetic proteins (BMPs). These changes were already evident at early stages of phenotypic alteration (Supplementary Fig. 10a), suggesting that augmented SMAD activation is not a secondary consequence of disrupted cerebrovascular morphogenesis. Moreover, the anatomical localization of vascular lesions coincides with areas of higher phospho-SMAD immunoreactivity (Supplementary Fig. 10b).

Analysis of single optical sections from confocal images revealed that increased SMAD3 phosphorylation is not restricted to pericytes of *Rbpj*iPC animals but also affects ECs, whereas SMAD1/5 phosphorylation was only detected in the mutant endothelium (Fig. 5f, g). Accordingly, sorted ECs from P7 mutant brain cortices showed a significant increase in the expression levels of *Id1* and *Id3* (Supplementary Fig. 10c), two well-characterized downstream targets of BMP-mediated signalling. Noteworthy, RNA-seq analysis of *Rbpj*iPC showed an upregulation of *Tgfb3* gene expression starting from P7 and aggravated at P10, a result that was confirmed by RT-qPCR interrogation of freshly sorted brain mural cells (Fig. 5h), and which is consistent with the increased phospho-SMAD3 levels seen both in mutant ECs and pericytes. At the same time, *Bmp2* and *Bmp4* transcripts were significantly augmented in freshly isolated P7 and P10 *Rbpj*iPC ECs from brain cortex (Fig. 5i), implying that the increased phosphorylation of SMAD1/5 is a consequence of excessive BMP production by ECs. Furthermore, endothelial expression of *Nrp1*, a co-receptor for vascular endothelial growth factor (VEGF) and semaphorins, which promotes tip cell function[32] and limits SMAD2/3 activation[33], was significantly downregulated in the *Rbpj*iPC brain vasculature (Fig. 5i).

It is well established that ECM proteins play a central role in shaping TGFβ and BMP signalling gradients by controlling the storage, localization, and activation of ligands[34, 35]. GSEA of consistently upregulated transcripts demonstrated a significant

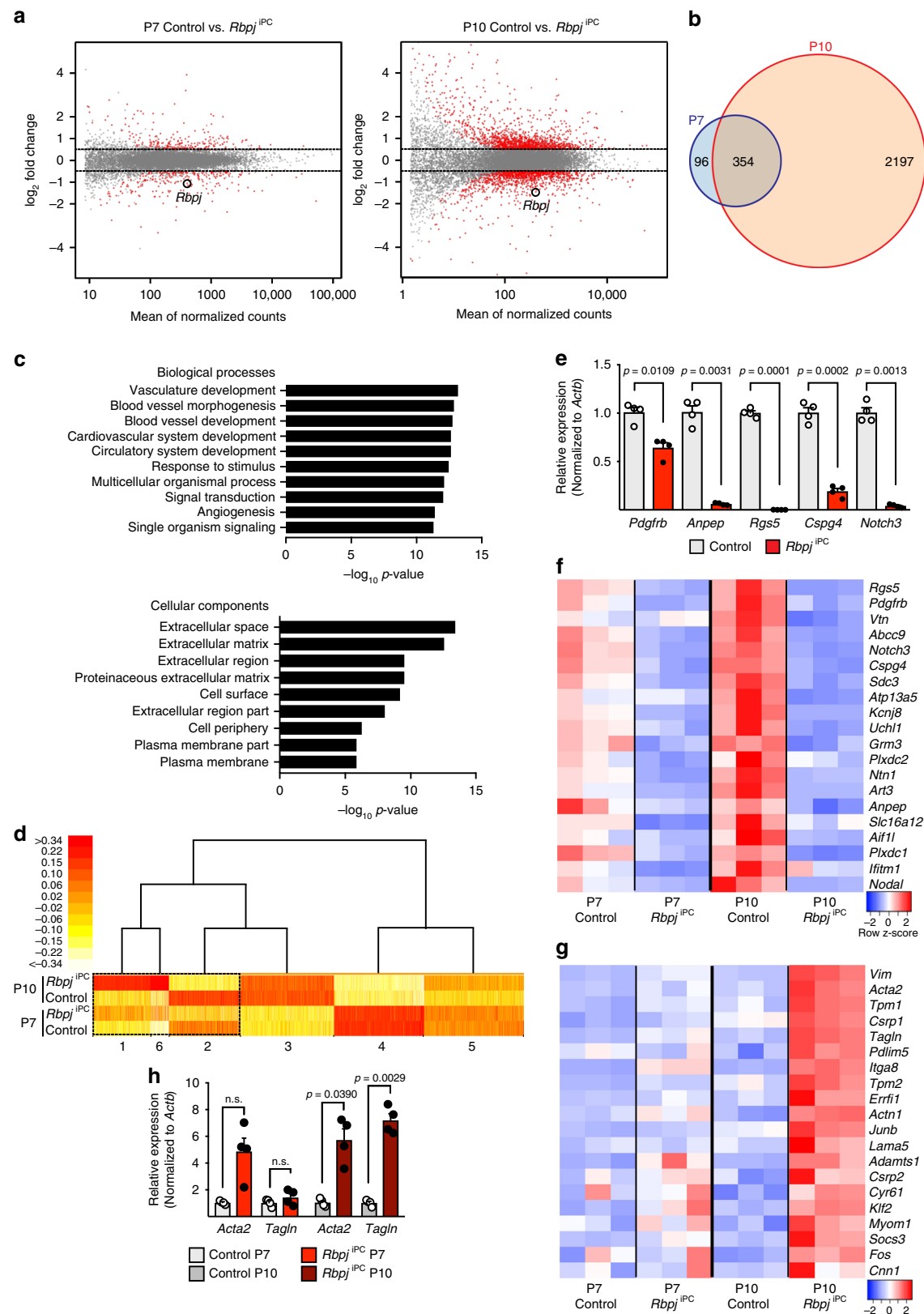

enrichment of genes encoding ECM and ECM-associated proteins including several TGFβ-related molecules in *Rbpj*iPC PDGFRβ⁺ cells (Supplementary Fig. 10d, e). Among these genes, matrix glycoproteins (i.e., Fibrillin-1, Fibrillin-2, Fibronectin, and Perlecan) responsible for binding, targeting, and concentrating the large latent complex of TGFβ in specific locations[36–38] were strongly upregulated in *Rbpj*iPC PDGFRβ⁺ cells (Fig. 5j), potentially facilitating the storage of latent TGFβ in the basement membrane shared by ECs and pericytes. At the same time, thrombospondin-1 (*Thbs1*), an important activator of latent TGFβ that disrupts the non-covalent interactions between the latency associated peptide (LAP) and mature TGFβ[34], was

**Fig. 3** Transcriptome profiling of *Rbpj*iPC mural cells. **a** MA-plots of DEGs between P7 and P10 control and *Rbpj*iPC mutant mural cells. The *x*-axis represents the mean normalized counts and the *y*-axis shows the log₂ fold change >0.5. Red dots correspond to statistically significant DEGs. *Rbpj* is indicated. **b** Venn diagram showing the overlap in DEGs between control and *Rbpj*iPC mutant mural cells at P7 and P10. **c** Top gene ontology (GO) biological process and cellular components terms related to differentially expressed genes in P7 *Rbpj*iPC mural cells (FDR, corrected *p*-value). **d** Model-based hierarchical clustering heat map of transcripts from control and *Rbpj*iPC mural cells at P7 and P10. Boxed region (dashed line) correspond to gene clusters which are consistently up (1 and 6) or downregulated (2) in *Rbpj*iPC mice both at P7 and P10 stages. **e** RT-qPCR analysis of putative pericyte markers in sorted mural cells from P10 control and *Rbpj*iPC brain cortices. *p*-values, Brown–Forsythe and Welch one-way ANOVA with Tamhane's T2 test, *n* = 4. **f** Heat map representation of known pericyte markers and recently proposed pericyte-enriched genes downregulated in *Rbpj*iPC mural cells at P7 and P10. **g** Heat map representation of arterial/arteriolar vascular SMC-enriched genes upregulated in *Rbpj*iPC mural cells at P7 and P10. **h** RT-qPCR analysis of vascular SMC markers in sorted mural cells from P7 and P10 control and *Rbpj*iPC brain cortices. *p*-values, Brown–Forsythe and Welch one-way ANOVA with Tamhane's T2 test, *n* = 4. Error bars represent s.e.m. Source data are provided as a Source Data file

upregulated, as confirmed by RNA-seq, qPCR of FACS-isolated mural cells, immunostaining, and in vitro after TAT-Cre-mediated *Rbpj* depletion (Fig. 5k, l).

TGFβ activation regulates ECM remodelling and has been associated with the induction of fibrosis by promoting the expression of basement membrane proteins, cell adhesion protein receptors, and metalloproteases as well as their inhibitors[39]. In this regard, several ECM proteins induced by TGFβ, such as plasminogen activator inhibitor 1 (*Serpine1*), connective tissue growth factor (*Ctgf*), TGFβ-induced (*Tgfbi*), tissue inhibitor of metalloproteinase 1 (*Timp1*), osteopontin (*Postn*), tenascin (*Tnc*), and matrix metalloproteinases (i.e., *Mmp10* and *Mmp25*), were upregulated in *Rbpj*iPC brain pericytes (Supplementary Fig. 10e, f). Likewise, strong changes in the expression pattern of integrin α subunits in pericytes together with increased activation of endothelial integrin β1 were obvious in the *Rbpj*iPC vasculature at P10, arguing further for extensive changes in cell–ECM interactions (Supplementary Fig. 11a–c). Importantly, no such changes were triggered by the inhibition of Notch signalling in brain mural cells (Supplementary Fig. 11d–g).

In order to address whether the gene expression changes associated to increased TGFβ signalling are mediated by direct binding of RBPJ, we performed ChIP-seq in cultured primary brain pericytes using an antibody targeted against the transcription factor (Supplementary Fig. 12a and Supplementary Table 1). As expected, the DNA binding motif for RBPJ was significantly enriched in the 11,094 peaks identified (Fig. 6a, Supplementary Fig. 12b, c and Supplementary Data 3). Moreover, GREAT analysis (Genomic Regions Enrichment of Annotations Tool) was able to identify the Notch signalling pathway as an over-represented functional term amongst the genes associated to RBPJ binding sites (Fig. 6b), further validating the specificity of the approach. In addition, ChIP-seq experiments using antibodies against the chromatin marks H3K4me3 and H3K4me1 revealed that RBPJ peaks are either enriched in promoters (H3K4me3⁺) or distal enhancer regions (H3K4me1⁺) of the bound genes (Fig. 6c). Among the RBPJ-bound genes, we identified 122 that are significantly upregulated in both P7 and P10 pups (Fig. 6d and Supplementary Data 4). Interestingly, this list includes several molecules that control TGFβ signalling (*Tgfb3*, *Thbs1*, *Fn1*, and *Fbn2*), drive fibrotic changes (*Tgfbi*, *Serpine1*) or are responsible for cell–ECM interactions (*Itga3*, *Itga5*, *Itga8*) (Fig. 6d and Supplementary Fig. 12d). In addition, we also found transcription factors, such as *Atf3* and *Runx1* (Fig. 6d and Supplementary Fig. 12d), which have been proposed to promote fibrosis in connection to increased TGFβ signalling[40, 41].

Altogether, our data shows that the loss of RBPJ in pericytes triggers a complex response involving excessive production of TGFβ3 and alterations in ECM composition, which lead to increased activation of SMAD3 together with non-cell-autonomous changes, such as increased phosphorylation of SMAD1/5, in the adjacent endothelium. Furthermore,

characterization of the genomic landscape in cultured pericytes revealed that RBPJ binds the transcriptional domains of genes controlling TGFβ signalling. Many of these genes are significantly overexpressed after the loss of *Rbpj* function, further suggesting a critical repressor role of RBPJ in pericytes during early postnatal development.

**Pathological implications of *Rbpj* inactivation in pericytes**. Tamoxifen administration over 3 consecutive days starting from P3 or P5 produced *Rbpj*iPC mutants with prominent vascular lesions in the cerebellum (Supplementary Fig. 13a), which recapitulate defects in the cerebellar white matter of mouse models for CCMs[42–44]. Common features include dilated, haemorrhagic venules in the white matter with secondary inflammatory reaction (Fig. 7a). Another hallmark of CCMs is the cleavage of the matrix proteoglycan versican by the metalloprotease ADAMTS-4, leading to the exposure of the neo-epitope DPEAAE[43]. Abluminal presence of DPEAAE was obvious in *Rbpj*iPC cerebellar lesions (Supplementary Fig. 13b). CCMs arise from mutations affecting the genes encoding KRIT1/CCM1, CCM2, or PDCD10/CCM3 proteins in the endothelium[44], but the corresponding transcripts were only slightly reduced in the *Rbpj*iPC brain endothelium (Fig. 7b). Activation of the MEKK3-MEK5-ERK5-MEF2 signalling axis leading to increased expression of the Kruppel-like family transcription factors Klf2 and Klf4 is a key process in CCM pathogenesis[43, 45]. In addition, integrin β1 activation has been reported as a signal triggering elevated *Klf2* expression[46]. Strikingly, *Klf2* and *Klf4* were significantly upregulated in the *Rbpj*iPC endothelium, and both Klf4 protein and activated integrin β1 were strongly increased in mutant cerebellar white matter lesions (Fig. 7b–e).

Next, we further evaluated the effect of *Rbpj* deletion in adult animals. Surprisingly, tamoxifen-induced recombination after weaning (3 weeks of age), did not induce any obvert phenotype in the CNS of *Rbpj*iPC mutants, which showed normal vascular organization, proper expression of mural cell differentiation markers, and no BBB defects (Supplementary Fig. 13c, d). Given that *Rbpj* deletion did not affect neurovascular homeostasis in this setting, we used the permanent distal middle cerebral artery occlusion (dMCAO) model to induce ischaemic stroke in the adult brain cortex. Angiogenesis is a primary neurovascular response during stroke recovery[47] and pericytes play important, yet not fully understood roles in different stages of ischaemic stroke[48]. As *Pdgfrb-CreERT2* transgenic mice enable efficient targeting of mural cells in adult mice, we first performed lineage tracing analysis in combination with the *Rosa26*ᵐᵀᵐᴳ Cre reporter allele in animals with one or two functional copies of *Rbpj*. Tamoxifen was administered 1 week before dMCAO surgery and samples were collected 7 days post-operation. As expected, the ischaemia-affected brain hemisphere showed strong neuronal depletion and dilation of vascular structures together with significant upregulation of PDGFRβ expression

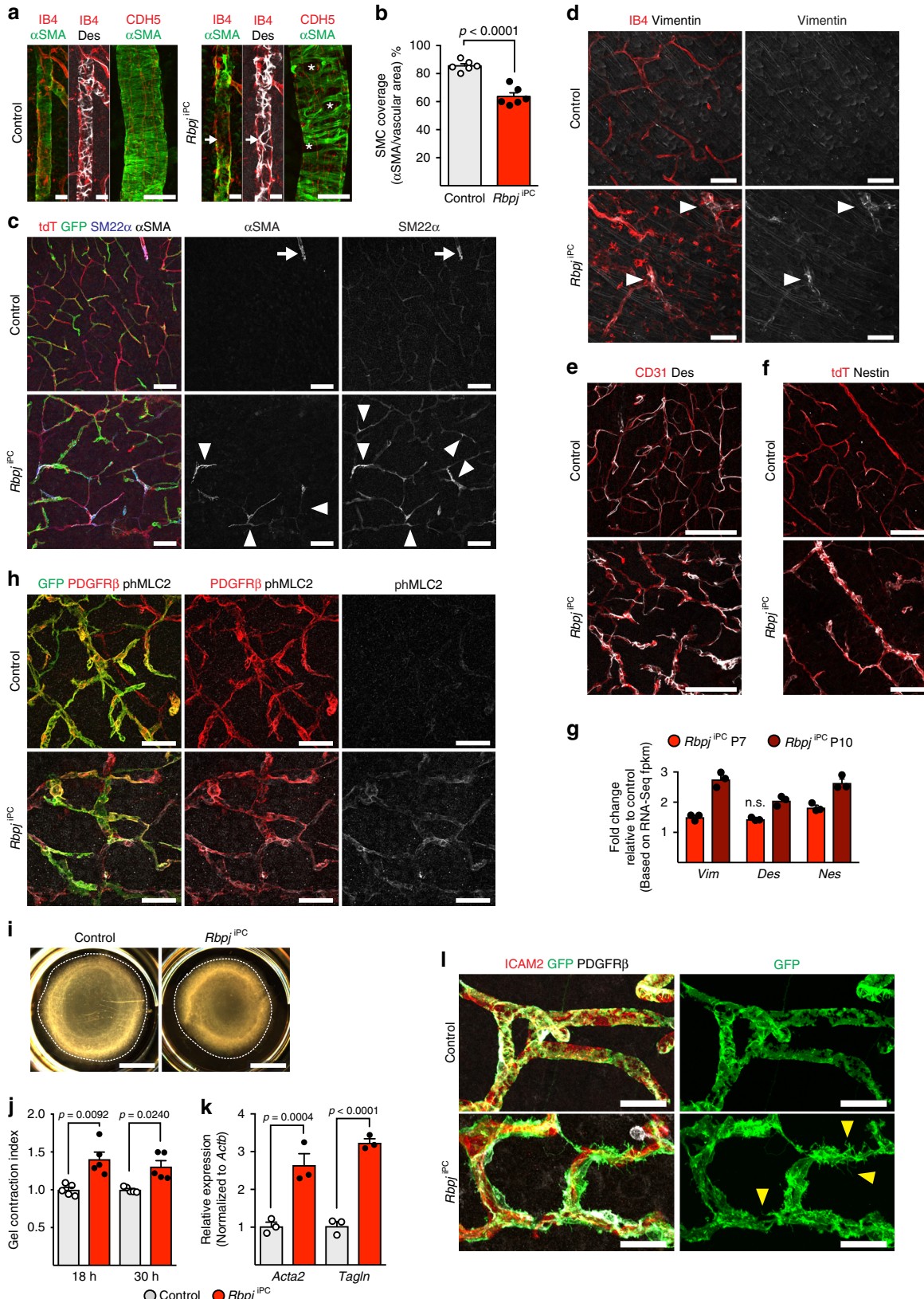

(Supplementary Fig. 13e). Interestingly, ischaemia-induced PDGFRβ expression was not restricted to perivascular locations and clearly exceeded the area covered by recombined (GFP⁺) pericytes (Fig. 7f), suggesting post-ischaemic recruitment of PDGFRβ⁺ cells or de novo expression of this receptor in previously negative cells. GFP-labelled pericytes showed drastic

(Supplementary Fig. 13e). Interestingly, ischaemia-induced PDGFRβ expression was not restricted to perivascular locations and clearly exceeded the area covered by recombined (GFP$^+$) pericytes (Fig. 7f), suggesting post-ischaemic recruitment of PDGFRβ$^+$ cells or de novo expression of this receptor in previously negative cells. GFP-labelled pericytes showed drastic

morphological changes in comparison to mural cells of the contralateral hemisphere. Typically, pericytes are tightly attached to ECs, but after stroke they emit long protrusions that bridge adjacent capillaries (Fig. 7f and Supplementary Fig. 13e, f).

In order to assess whether *Rbpj* is necessary for pericytes during the vascular response to ischaemic stroke, adult control

**Fig. 4** Functional defects of *Rbpj* mutant mural cells. **a** Confocal images of P10 cortical arteries stained for isolectin B4 (IB4) or CDH5 (red), αSMA (green), and Desmin (Des, white). Note irregular SMC coverage of mutant arteries (asterisk) and lack of αSMA labelling in Des$^+$ regions (white arrow). Scale bar, 50 μm (left panels) and 25 μm (right panel, high magnification). **b** SMC coverage in P10 brain cortex arteries. *p*-values, Student's *t*-test, *n* = 6. **c** Confocal images of P10 cortical vasculature (tdT$^+$, red) stained for GFP (green), SM22α (blue, white), and αSMA (white). Note the expression of αSMA and SM22α in the *Rbpj*$^{iPC}$ microvasculature (white arrowheads). Scale bar, 100 μm. **d**–**f** Confocal images of brain cortex vasculature showing increased mural cell-specific staining of Vimentin (**d**, arrowheads), Desmin (Des, **e**), and Nestin (**f**) (all in white) in P10 *Rbpj*$^{iPC}$ blood vessels (labelled with IB4, CD31, or tdT, red) relative to control. Scale bar, 50 μm (**d**) or 100 μm (**e**, **f**). **g** Gene expression fold change (RNA-seq fpkm) in *Rbpj*$^{iPC}$ mutants relative to age-matched controls (set as 1). *p*-adjusted values < 0.0001 for all with exception of *Des* expression at P7. *n* = 3. n.s., not statistically significant. **h** Confocal images of P10 cortical blood vessels showing increased immunostaining for phospho-myosin light chain 2 (phMLC2, white) in pericytes (PDGFRβ$^+$, red; GFP$^+$, green). Scale bar, 50 μm. **i** Collagen gel contraction assay showing increased contractile ability of lentivirus-Cre transfected *Rbpj*$^{iPC}$ relative to control primary mouse brain pericytes. Scale bar, 4 mm. **j** Contractility quantitation at 18 and 30 h after pericyte seeding. *p*-values, Kruskal–Wallis with Dunn's test, *n* = 5. **k** RT-qPCR analysis of *Acta2* and *Tagln* expression in cultured pericytes. *p*-values, one-way ANOVA with Sidak's test, *n* = 3. **l** Confocal images of P10 brain cortex microvasculature stained for ICAM2 (red), GFP (green), and PDGFRβ (white). Cre-induced membrane-tagged GFP expression allows detailed imaging of abnormal protrusions emerging from *Rbpj*$^{iPC}$ pericytes (yellow arrowheads). Scale bar, 25 μm. Error bars represent s.e.m. Source data are provided as a Source Data file

and *Rbpj*$^{iPC}$ mice were subjected to dMCAO. *Rbpj*$^{iPC}$ mutants developed bigger cortical lesions with larger space-occupying effect due to brain oedema and increased corrected lesion volumes (Fig. 7g, h). Histologic analysis revealed higher phospho-SMAD3 immunoreactivity in *Rbpj*$^{iPC}$ than in control stroke samples (Fig. 7i), an effect that partially resembles the phenotype seen in postnatal mutants. In addition, a stronger inflammatory response was detected in *Rbpj*$^{iPC}$ animals together with localized vascular aberrations such as vessel tortuosity and ballooning (Fig. 7i, j). Pericytes have been proposed to have an important role in neuroinflammation, mostly through secretion of cytokines[49]. In this regard, our RNA-seq analysis of mutant pericytes from young pups revealed strong overexpression of pro-inflammatory chemokines (Fig. 7k) together with upregulation of inflammation-related adhesion molecules, MMPs, and phagocytic receptors (Supplementary Fig. 13g). Interestingly, (C–C-motif) ligand 2 (*Ccl2*) is induced upon *Rbpj* deletion since early stages (P5 and P7), is strongly upregulated shortly after *Rbpj* deletion in vitro (Supplementary Fig. 13h), and is bound by RBPJ (Fig. 7l), suggesting a potential direct role of RBPJ in controlling a pro-inflammatory genetic program in pericytes, which, if activated, is likely to worsen stroke outcome.

## Discussion

The critical role of pericytes in BBB development[50] and maintenance[4] is well appreciated and, conversely, pericyte loss can trigger brain vascular damage that ultimately leads to secondary neurodegenerative changes[8]. Altogether, the current view of pericytes portraits them as protective cells that safeguard neurovascular homeostasis. Here, we provide evidence that the loss of RBPJ generates dysfunctional pericytes that compromise neurovascular homeostasis and induce severe vascular lesions that are not present in animals with acute or chronic pericyte depletion. *Rbpj*-deficient pericytes induce pathogenic transformation of the vasculature resembling CCMs at the morphological and molecular level, and result in bigger stroke lesions upon ischaemic insult.

Single-cell molecular profiling of brain mural cells has suggested that pericytes and vein-associated SMCs represent a continuum that is distinct from the SMCs covering arteries and arterioles[24]. Consistent with this finding, *Pdgfrb-CreERT2*-mediated *Rbpj* inactivation prominently affects the morphogenesis and integrity of capillaries and veins, and drives a profound change in mural cell identity characterized by reduced expression of pericyte-specific markers and upregulation of genes characteristic for arterial SMCs. This indicates that proper control of the molecular and functional properties of pericytes is critical for correct vascular patterning, function, and BBB integrity. Moreover, our data indicate that RBPJ functions in a Notch-independent fashion in postnatal brain

pericytes. Previous reports highlighting activation of Notch targets such as *Hey1* in retinal pericytes associated to the retinal angiogenic front[12] may reflect tissue-specific programs, alternative signalling mechanisms driving *Hey1* expression, or differential control of pericyte behaviour in the context of angiogenic growth into avascular (retinal) tissue.

Based on our in vivo data, we also propose that RBPJ helps to repress SMC differentiation, which is consistent with a recent in vitro study showing that *RBPJ* depletion in human aortic SMCs enhances the differentiation of these cells in a similar way as TGFβ1 stimulation[51]. The same study also found that *RBPJ* knockdown results in increased association of active transcription marks with a large number of TGFβ signalling and matrix-associated genes, which is consistent with our own results. Apart from enhancing vascular SMC differentiation and contractility, TGFβ signalling has been implicated in the pathogenesis of tissue fibrosis[39] and CCMs[44], and may induce a pro-inflammatory state in pericytes through the expression of chemokines[49]. In addition, excessive activation of SMAD2/3 in ECs has been shown to limit tip cell specification[33], potentially explaining the strong reduction of sprouting angiogenesis in *Rbpj*$^{iPC}$ mutants. Moreover, *Rbpj*-mutant pericytes may directly promote inflammatory processes in the brain by excessive production of CCL2, an important mural cell-derived mediator of neuroinflammation[52]. Interestingly, BMP expression in ECs is increased after exposure to proinflammatory stimuli[53] and once up-regulated, BMP2 and BMP4 induce pro-inflammatory effects[54] and activation of SMAD1/5, which has been shown to drive cell cycle progression leading to excessive cell proliferation[55].

The proposed Notch-independent roles of RBPJ in pericyte biology are particularly important during scenarios where physiological or regenerative angiogenesis is involved, presumably reflecting its relevance for fate commitment and intercellular communication in environments associated to increased cellular plasticity. Moreover, the severe defects on vascular integrity induced by mural cell-specific deletion of *Rbpj* strongly argue for potential involvement of pericytes in pathogenic transformation. On this regard, it should be noted that the molecular mechanisms responsible for CCM lesions development (namely increased RhoA activity[56], augmented MEKK3/KLF2,4 signalling[43], and abnormal activation of the TGFβ/BMP/SMAD pathway[44]) may be influenced by additional mechanisms such as the behaviour of pericytes. Likewise, the recently appreciated relevance of mural cells in the elicitation of neuroinflammation[52] further highlights the important implications of pericytes in pathogenic responses and recovery after insult.

Based on the sum of our findings, we propose that improper intercellular communication elicited by dysfunctional pericytes

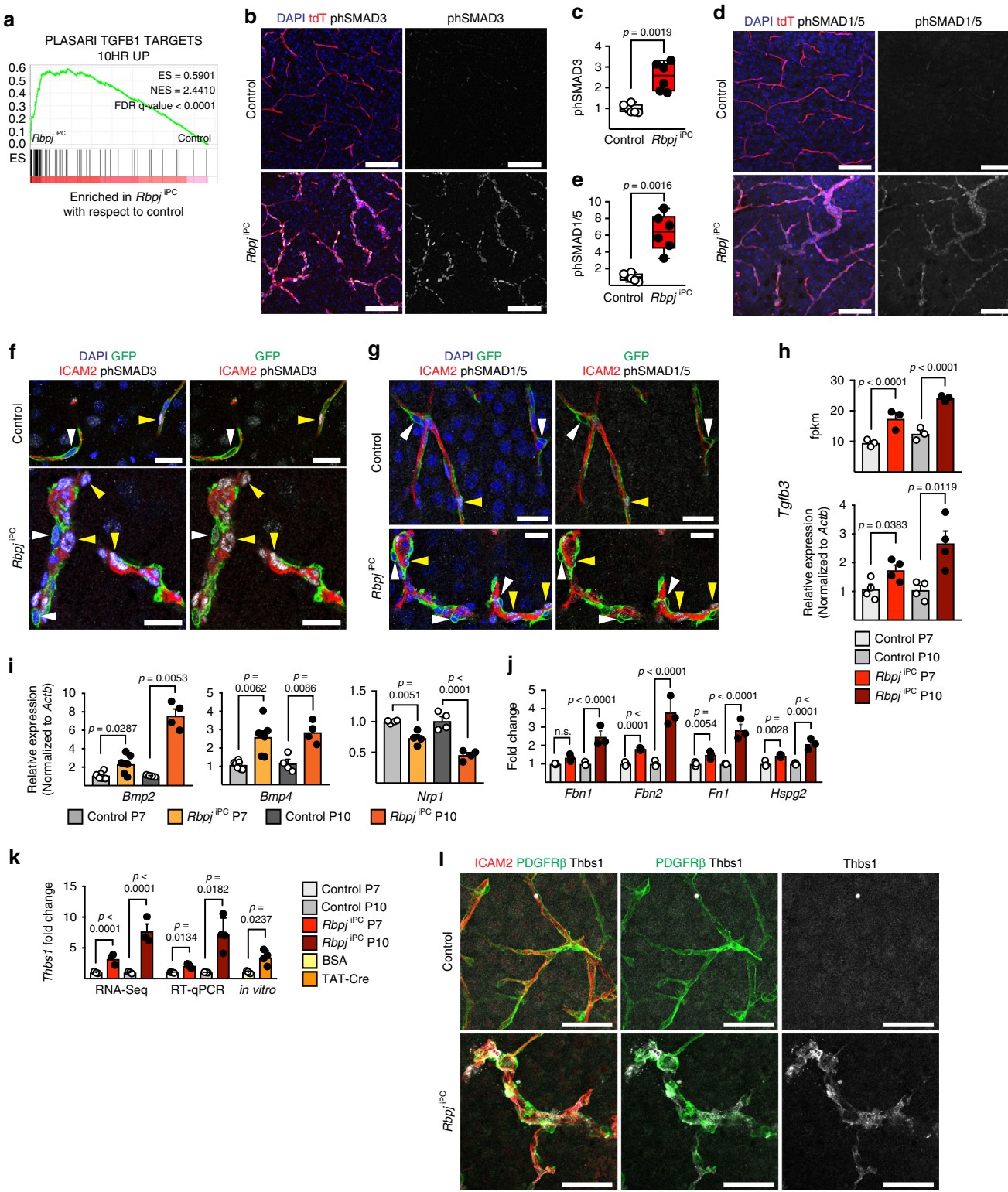

can lead to profound changes in the adjacent endothelium and the surrounding tissue, which together drive neurovascular pathogenesis.

## Methods

**Mutant mouse models and inducible genetic experiments**. *Pdgfrb-CreERT2*[12] were interbred with *Rosa26*[mTmG] Cre-reporter mice[13] in order to monitor recombination efficiency and specificity. Mural cell-specific *Rbpj* conditional knockouts (*Rbpj*[iPC]) were generated by crossing *Pdgfrb-CreERT2* with *Rbpj*[lox]

mice[14]. For some experiments the *Rosa26*[mTmG] reporter allele was included in the *Pdgfrb-CreERT2 Rbpj*[lox] background. In such cases, *Pdgfrb-CreERT2*[+/T] *Rbpj*[l+/lox] double heterozygous animals were used as controls, whereas Cre-negative littermates were used as controls in other experiments. Animals with a single-allele inactivation of *Rbpj* were indistinguishable from Cre-negative littermates.

For quantitative transcriptome analysis, *Pdgfrb-CreERT2 Rbpj*[lox/lox] mice were interbred with *Rpll22*[tm1.1Psam] (RiboTag) animals[22]. As control for RNA-seq analysis, age and weight-matched pups bearing the *Pdgfrb-CreERT2* transgene and two copies of the RiboTag allele, but wildtype for *Rbpj*, were used.

**Fig. 5** Increased TGFβ signalling in the $Rbpj^{iPC}$ cerebral vasculature. **a** GSEA of overexpressed genes (RNA-seq model-based clustering analysis) reveals upregulation of TGFβ targets in $Rbpj^{iPC}$. ES, enrichment score; NES, normalized enrichment score; FDR, false discovery rate. **b** Confocal images showing increased phosphorylation of SMAD3 (phSMAD3, white) in P10 $Rbpj^{iPC}$ brain cortex vasculature (tdT+, red). Scale bar, 100 μm. **c** phSMAD3 immunosignal intensity quantitation in P10 brain vasculature. p-values, Welch's t-test, n = 6. **d** Confocal images showing increased phosphorylation of SMAD1/5 (phSMAD1/5, white) in P10 $Rbpj^{iPC}$ brain cortex vasculature (tdT+, red). Scale bar, 100 μm. **e** phSMAD1/5 immunosignal intensity quantitation in P10 brain vasculature. p-values, Welch's t-test, n = 6. **f, g** Confocal single optical sections showing localization of phSMAD3 (**f**) or phSMAD1/5 (**g**) (white) in the nuclei (DAPI+, blue) of ECs (ICAM2+, red; yellow arrowheads) or pericytes (GFP+, green; white arrowheads) in P10 cortical vessels. Note phSMAD3 presence in ECs and pericytes, while phSMAD1/5 is only detectable in ECs. Scale bar, 25 μm. **h** $Tgfb3$ expression in cortical mural cells from P7 or P10 mice analyzed by RNA-seq (top) or RT-qPCR (bottom). p-adjusted values (RNA-seq) or p-values, Student's t-test, n = 3–4. **i** RT-qPCR analysis in sorted ECs from P7 and P10 brain cortices. p-values, Brown–Forsythe and Welch one-way ANOVA with Tamhane's T2 test ($Bmp2$ and $Bmp4$), or one-way ANOVA with Sidak's ($Nrp1$), n = 4–7. **j** Gene expression fold change based on RNA-seq counts in P7 and P10 brain cortex mural cells. p-adjusted values. n = 3. **k** Thrombospondin-1 ($Thbs1$) expression analysis in vivo (RNA-seq and RT-qPCR for FACS-sorted mural cells) and in vitro in cultured pericytes. p-adjusted values (RNA-seq), p-values, Welch's t-test (in vitro), or Brown–Forsythe and Welch one-way ANOVA with Holm–Sidak's test (RT-qPCR), n = 3–4. **l** Confocal images showing increased expression of thrombospondin-1 (Thbs1, white) in mural cells (PDGFRβ+, green) of P10 $Rbpj^{iPC}$ cortical vasculature (ICAM2+, red). Scale bar, 50 μm. Error bars represent s.e.m. Source data are provided as a Source Data file

For acute mural cell-ablation experiments, $Pdgfrb$-CreERT2 mice were interbred with $Gt(ROSA)26Sor^{tm1(DRA)Jpmb}$/J ($ROSA$-DTA) knockin mice[18]. Chronic paucity of mural cell coverage was analyzed in $Pdgfb^{Ret/Ret}$ animals[19].

Analysis of the activation of Notch1 in mural cells was performed in $Notch1^{tm3(cre)Rko}$/J mice[28] interbred with $Rosa26$-mTmG mice[13]. For Notch loss-of-function analysis, constitutive knockout mice for $Notch3$ (B6:129S1-$Notch3^{tm1Grid}$/J[26]) were interbred with mice bearing $Notch2$ floxed alleles[57] and the $Pdgfrb$-CreERT2 transgene. Alternatively, the $Pdgfrb$-CreERT2 mouse model was crossed with $Rosa26^{dnMaml1}$ mice[29] to induce conditional expression of dominant negative Mastermind-like 1.

For inducing constitutive canonical Notch signalling in mural cells, the $Pdgfrb$-CreERT2 mouse line was bred with $Gt(ROSA)26Sor^{tm1(Notch1)Dam}$ mice[30] in which the Rosa26 locus drives expression of the Notch1 intracellular domain upon Cre-mediated recombination.

Analysis of canonical Notch signalling activation in mural cells was performed in mice heterozygous for the reporter alleles Hey1-EGFP ($Tg(Hey1$-EGFP$)$ $ID40Gsat$) and CBF:H2B-Venus ($Tg(Cp$-HIST1H2BB/Venus$)47Hadj$/J)[25].

Whenever Cre-mediated recombination was induced in young pups, 50 μg of tamoxifen (Sigma, T5648; dissolved in ethanol-peanut oil (Sigma, P2144) 1:40 at 1 mg ml$^{-1}$) were intraperitoneally (i.p.) injected every 24 h during 3 consecutive days starting at postnatal day 1 (P1). Pups were analyzed at specific postnatal stages, as described for each experiment and corresponding figure legend. Tamoxifen injection to juvenile or adult animals (3 weeks and older) consisted of i. p. injections of 1 mg tamoxifen (1 mg ml$^{-1}$) over 5 consecutive days. For inducing Cre-recombinase activity during embryonic development, 3 mg of tamoxifen (10 mg ml$^{-1}$) were injected i.p. to pregnant females at E11.5 (taking the day of vaginal plug identification as E0.5) and embryos were collected at E14.5.

In all genetic experiments performed with young pups, both females and males were used in a 1:1 ratio. The permanent dMCAO ischaemic stroke model was performed in >20-week-old adult male mice.

All experiments involving animals were performed in compliance with the relevant laws and institutional guidelines, following protocols previously approved by local animal ethics committees and conducted with permission granted by the Landesamt für Natur, Umwelt und Verbraucherschutz (LANUV) of North Rhine-Westfalia to the Max Planck Institute of Molecular Biomedicine and by the Landesamt für Gesundheit und Soziales (G0298/13) of Berlin to the Center for Stroke-Research Berlin (CSB), Charité-Universitätsmedizin.

**Brain sample preparation and immunohistochemistry.** For immunohisto-chemical analysis of mouse brains, pups at postnatal day (P)7, 10 or adult mice were anaesthetized by intraperitoneal injection of xylazine (Bayer, Rompun 2%; 10 mg kg$^{-1}$) and ketamine (Zoetis, Ketavet 100 mg ml$^{-1}$; 100 mg kg$^{-1}$) dissolved in saline. After opening the thoracic cavity and exposing the heart, the right atrium was pierced and 10 ml (20 ml for adults) of PBS were perfused through the left ventricle using a peristaltic pump (Pump P-1, GE Healthcare) in order to wash out the blood from the major circulatory system. Right after, 10 ml (20 ml for adults) of ice-cold 4% paraformaldehyde (PFA, Sigma, P6148) were perfused in the same manner to start fixation of the tissues. Brains were carefully dissected out from the skull and postfixed for up to 8 h by immersion in 4% PFA at 4 °C, then washed in PBS and sectioned using an acrylic brain matrix for mouse (RBMA-200C, World Precision Instruments) either through the sagittal mid-line or in 2-mm thickens coronal sections. Brain hemispheres or coronal sections were embedded in 55 °C low gelling temperature agarose (Sigma, A9414, 4% in PBS) and immediately placed on top of ice to cool down. Once solidified, agarose blocks were trimmed, glued to a specimen holder with cyanoacrylate (UHU GmbH & Co. KG) and 100 μm sections were obtained using a vibratome (VT 1200S, Leica). Vibratome sections were blocked and permeabilized by overnight (ON) incubation in 1% BSA (Sigma, P6148) and 0.5% Triton-X-100 (Sigma, T8787) in PBS at 4 °C. Primary antibodies were diluted in blocking/permeabi-lization solution supplemented with 2% normal donkey serum (Abcam, ab7475) and incubated overnight (ON) at 4 °C. After staining with primary antibodies, vibratome sections were washed once in 0.5% Triton-X-100 in PBS and three times in PBS (20 min at 4 °C each) and incubated ON with suitable donkey-raised, species-specific, Alexa Fluor-coupled secondary antibodies (all from Invitrogen) diluted (1:400) in blocking/permeabilization buffer with 2% normal donkey serum. After secondary antibody incubation, vibratome sections were washed as already described before mounting in Fluoromount G (Southern Biotech, 0100-01).

Alternatively, for specific antibodies, in vivo perfusion was performed with ice-cold 1% PFA as described and brains were fixed ON in methanol (4 °C). The next day, samples were transferred to fresh methanol (−20 °C) and stored. Before agarose embedding, brains fixed in methanol were incubated for 15–20 min in increasing concentrations of PBS:methanol (25, 50, 75, and 100% PBS) to rehydrate the tissues. After final incubation in PBS, brain samples were sectioned and embedded in agarose as described.

The following primary antibodies were used for immunostaining: rabbit anti-AQP4 (1:200, Sigma HPA014784), mouse anti-αSMA-Cy3 (1:300, Sigma C6198), mouse anti-αSMA-660 (1:300, eBioscience 50-9760-82), rat anti-CD13 (1:100, abD Serotec, MCA2183GA), goat anti CD31 (1:200, R&D Systems, AF3628), rat anti VE-cadherin (CDH5, 1:100, BD Biosciences, 555289), rabbit anti-Claudin 5 (CLDN5, 1:100, Thermo Fisher Scientific, 34-1600), rabbit anti-cleaved Caspase-3 (1:100, Cell Signaling, 9664), rabbit anti-Desmin (1:100, Abcam, ab15200), rabbit anti DPEAAE (1:100, Thermo Scientific, PA1-1748A), rabbit anti ERG (1:100, Abcam, ab110639), rabbit anti-GFAP (1:200, DAO, Z0334), chicken anti-GFP (1:300, 2BScientific Ltd., GFP-1010), rabbit anti-GLUT1 (1:200, Millipore, 07-1401), goat anti-Iba1 (1:100, Novus Biologicals, NB100-1028), rat anti-ICAM2 (1:200, BD Pharmingen, 553326), goat anti-integrin α8 (Itgα8, 1:100, Thermo Fisher Scientific, FA5-47572), goat anti-KLF4 (1:100, R&D Systems, AF3158), rabbit anti MAP-2 (1:100, Abcam, ab32454), rat anti-nestin (1:200, Santa Cruz, sc101541), rabbit anti-NeuN (1:200, Cell Signalling, 12943), chicken anti-neurofilament H (NF-H, 1:100, Neuromics, CH22104), rabbit anti-NG2 (1:100, Millipore, AB5320), goat anti-Notch3 (1:100, R&D Systems, AF1308), rat anti-PDGFRβ (1:100, eBioscience, 14-1402), goat anti-PDGFR PDGFRβ (1:100, R&D Systems, AF1042), rabbit anti-phospho myosin light chain 2 (phMLC2, 1:100, Cell Signalling, 3674), rabbit anti-phosphoSMAD3 (phSMAD3, 1:100, Abcam, ab52903), rabbit anti-phospho SMAD1/5 (phSMAD1/5, 1:100, Cell Signalling, 9516), goat anti-podocalyxin (PODXL, 1:200, R&D Systems, AF1556), rabbit anti SM22α (1:200, Abcam, ab14106), rat anti-TER-119 (1:200, R&D Systems, MAB1125), rabbit anti-thrombospondin 1 (Thbs1, 1:100, Abcam, ab85762), goat anti-VEGR2 (1:100, R&D, AF644), rabbit anti-vimentin (1:100, Abcam, ab7783) and rabbit anti-zona occludens 1 (ZO1, 1:100, Invitrogen, 402200). In addition, biotin-conjugated isolectin B4 (IB4, 1:25, Vector Laboratories, B-1205) was used for visualization of blood vessels and inflammatory cells.

The following donkey-raised secondary antibodies (all in 1:400 dilution unless otherwise stated) were used for immunostaining: anti-rabbit IgG conjugated to Alexa Fluor (AF) 488 (Thermo Fisher Scientific, A21206), anti-chicken IgY AF488 (Jackson ImmunoResearch, 703-545-155), anti-rat AF488 (Thermo Fisher Scientific, A21208), anti-goat IgG AF488 (Invitrogen, A-11055), anti-rat IgG Cy3 (Jackson ImmunoResearch, 712-165-153), anti-rabbit IgG AF546 (Thermo Fisher Scientific, A10040), anti-goat IgG AF546 (Invitrogen, A-11056), anti-rat IgG AF594 (Thermo Fisher, A21209), anti-rabbit IgG AF594 (Thermo Fisher Scientific, A21207), anti-goat IgG AF594 (Thermo Fisher Scientific, A-11058), anti-rabbit IgG AF647 (Thermo Fisher Scientific, A31573), anti-rat IgG AF647 (Jackson ImmunoResearch, 712-605-153), and anti-goat IgG AF647 (Thermo Fisher

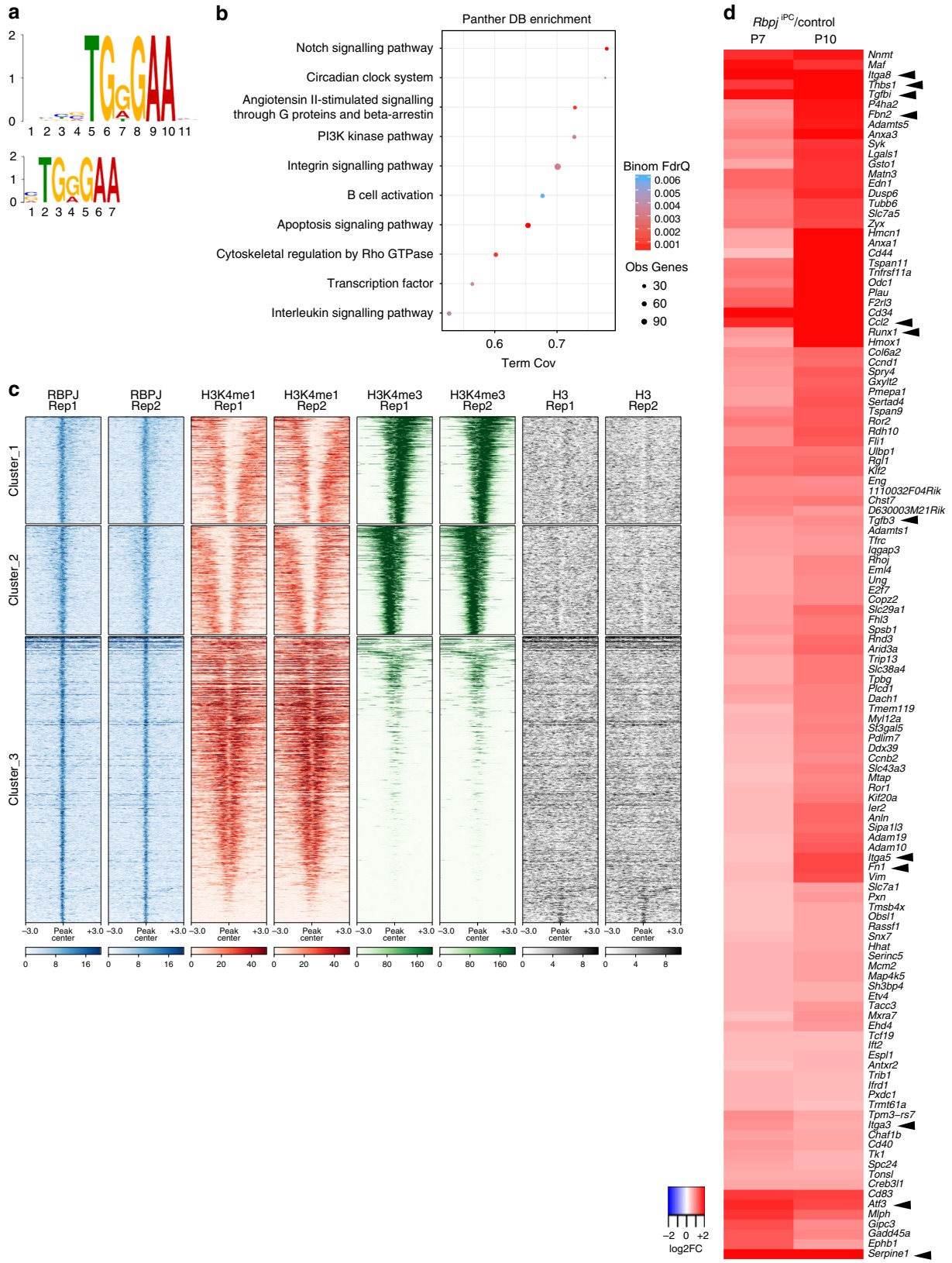

Scientific, A21447). Streptavidin AF405 (1:200, Invitrogen S32351) was used for detection of biotinilated-IB4 stained samples. Nuclei were counterstained with 4′,6-diamidino-2-phenylindole (DAPI, Sigma, D9542) diluted at 1 μg ml$^{-1}$ together with the secondary antibodies.

Processing of samples and staining conditions were kept identical and performed in parallel on samples from mutant mice and their respective controls for each experiment.

**Proliferation assay in vivo**. For in vivo analysis of cell proliferation, pups were injected intraperitoneally with 5-ethynyl-2-deoxyuridine (EdU, Life Technologies, A10044; 30 mg kg$^{-1}$ dissolved in DMSO-PBS 1:10 at 2 mg ml$^{-1}$) 4 h before dissection. Brains were collected and processed as described before. After secondary antibody staining, EdU labelling was detected by means of a Click-it EdU Alexa Fluor-647 Imaging Kit (Life Technologies, C10340) according to the manufacturer's instructions.

**Fig. 6** Characterization of the RBPJ-bound genomic landscape in cultured pericytes. **a** MEME (top) and DREME (bottom) analysis of the RBPJ-bound sites in pericytes identify the RBPJ binding motif amongst the top enriched motifs. **b** RBPJ binding sites were used for the identification of over-represented functional terms associated with RBPJ-bound genes. Depicted are the results for the most strongly enriched terms of the PantherDB database. The bubble plot encodes the false discovery rate corrected $p$-value (Binom FdrQ), the number of bound genes (Obs Genes), and the fraction of the total number of genes belonging to the corresponding term (Term Cov.). **c** RBPJ binding in pericytes occurs at both proximal (H3K4me3 positive clusters 1–2) and distal enhancer sites (H3K4me1 positive cluster 3). ChIP-seq was performed against RBPJ, H3K4me1, H3K4me3, and H3 in pericytes and RBPJ binding sites were clustered based on the differential enrichment of H3K4me1, H3K4me3, and nucleosome occupancy as revealed by using a panH3 antibody with k-means. Two replicates (Rep1 and Rep2) of each experiment are shown. **d** Identification of the genes bound by RBPJ and deregulated upon *Rbpj*-deletion in pericytes. Significantly up-regulated RBPJ-bound genes identified in P7 and P10 *Rbpj*[iPC] versus respective controls were selected based on adjusted $p$-value < 0.05 and log₂-transformed fold change > 0.5. Log-2 transformed transcriptional changes are shown as heatmap after hierarchical clustering using Euclidean distance measure and complete linkage. Black arrowheads point to relevant genes involved in increased TGFβ signalling and related ECM composition (*Tgfb3, Thbs1, Fbn2, Fn1*), cell–ECM crosstalk (*Itga8, Itga5, Itga3*), inflammation (*Ccl2*), and fibrosis (*Tgfbi, Serpine1, Runx1, Atf3*). Source data are provided as a Source Data file

**Image acquisition and quantitative analysis**. Whole-brain pictures and overview images from entire, unstained coronal slices were taken using a dissection microscope (Leica M205C) coupled to a digital camera (QImaging MicroPublisher 5.0 RTV) with Volocity 6.3 (Perkin Elmer) software.

Confocal images were acquired with a Zeiss LSM780 or LSM880 confocal laser scanning microscope equipped with the following objective lenses: 10× Plan APO NA 0.45, 20× Plan APO NA 0.80, water immersion 40× LD C APOchromat NA 1.10, and oil immersion 63× Plan APO NA 1.40. Alternatively, a Leica TCS SP8 confocal laser scanning microscope was used, equipped with the following objective lenses: 10× HC PL Apo NA 0.40, 20× HC PL APO CS2 NA 0.75, water immersion 40× HC PL APO W motCORR CS2 NA 1.10, and glycerol immersion 63× HC PL APO Glyc Corr CS2 NA 1.30. Image acquisition was done with the software ZEN software 2.3 SP1 FP1 14.0 (Carl Zeiss) or Leica Application Suite X 2.0.1.14392 (Leica). Images analysis and processing (brightness and contrast correction, background subtraction) was performed using Volocity 6.3 (Perkin Elmer), Photoshop CS6 13.0 (Adobe), Illustrator CS6 16.0.0 (Adobe), and Fiji-IJ 2.0.0-rc-43[58]. All confocal images are represented as maximum intensity projections unless stated otherwise.

For quantification of vascular area, the blood vessel markers CD31 or GLUT1 were used. Based on their expression pattern, objects were defined with the Find Objects tool from the Measurements package in Volocity and their area (μm²) measured. Vascular density was calculated by dividing the total perimeter of vessels identified with the luminal marker ICAM2 into the correspondent area calculated as already described. Vessel diameter of arteries and veins was measured manually in Fiji from CD31 stained vibratome sections. Arteries and veins were identified based on their morphology and the distinctive coverage of SMC (αSMA⁺). Capillary diameter was measured using a specific plugin[59] in ImageJ. Details of number of vessels measured per animal and condition can be found in the Source Data File.

Quantification of sprouts was performed in GLUT1 stained vibratome sections. Visual identification of sprouts was based in the identification of vascular projections with obvious filopodia leading their growth into the avascular space. The number of sprouts was normalized to the vascular area present in each image quantified.

Red blood cell (RBC) leakage was measured manually by defining the extravascular areas in the brain parenchyma where RBCs (identified by staining with GLUT1) were obvious and normalizing it to the whole brain region imaged. Percentage (%) values represent the proportion of the brain area in which extravascular RBCs are found.

For quantification of EC numbers combination of a vascular marker together with the EC-specific nuclear marker ERG was used for unambiguous identification. EC density reflects the number of ECs found in 100 μm² of vascular area. Proliferating ECs (EdU⁺ and ERG⁺) were visually identified, manually counted and their number normalized to the total amount of ECs in 100 μm² of vascular area.

Analysis of apoptosis in brain ECs was based on the identification of apoptotic ECs either by terminal deoxynucleotidyl transferase dUTP nick end labelling using the Click-iT TUNEL Alexa Fluor 647 Imaging Assay (Thermo Fischer Scientific, C10247) or by immunostaining against cleaved Caspase-3. The number of apoptotic ECs per 100 μm² of vascular area (measured as already described) was plotted.

Mural cell coverage was calculated by dividing the surface area positive for either pericyte (PDGFRβ) or SMC (αSMA) markers into the surface area stained for vascular markers (i.e., CD31 or GLUT1). Likewise, astrocyte endfeet coverage was calculated by dividing the surface area positive for Aqp4 immunostaining into the total vascular area. Recombination efficiency in mural cells was quantified by establishing a percentage relation between the area covered by recombined (GFP⁺) pericytes with regards to the total perivascular PDGFRβ⁺ area.

GFAP and Iba1⁺ regions were identified with the Find Objects tool from the Measurements package in Volocity and their areas (μm²) measured. Values are reported as such without additional normalization. The size of the fields of view used for quantitation was equal for control and mutant mice.

For quantitation of phSMAD3, phSMAD1/5, and KLF4 immunoreactivity signal intensity, the vascular area was identified and segmented based on blood vessel-specific staining (i.e., ICAM2) and tdTomato (tdT) labelling (in animals bearing the *Rosa26*^mTmG Cre-reporter allele). Within this area, DAPI-stained nuclei were identified and further segmented as region of interest (ROI). The mean signal intensity for the antibody of interest was calculated within the ROI and the average background signal intensity was subtracted. For calculating the background signal, the signal intensity in the vascular area after cropping out the DAPI⁺ nuclei (where the specific staining is expected) was measured. Active Itgβ1 staining quantitation was based in measurement of the signal intensity within the vascular area normalized to the non-vascular regions of the image. All signal intensity plots correspond to the value obtained after subtracting the background-specific mean intensity from the staining-specific mean intensity for each antibody and image quantified.

Detailed information regarding the number of biological replicates (animals or individual experiments), and the number of samples measured for each animal (or technical replicates in each experiment) are detailed in the Source Data File. Settings for scanner confocal detection and laser excitation were always kept identical between samples whenever comparisons between mutant mice and their respective controls were done.

**Blood flow assessment using two-photon in vivo microscopy**. P10 pups were anaesthetized by intraperitoneal injection of xylazine (Bayer, Rompun 2%; 3.3 mg kg⁻¹) and ketamine (Zoetis, Ketavet 100 mg ml⁻¹; 33.3 mg kg⁻¹) dissolved in saline and kept thereafter in a heating pad at 37 °C. Once the stage of surgical anaesthesia was achieved, a single midline incision on the skin of the head was performed and a custom-made titanium ring was glued on top of the intact skull. Next, 25 μL of Texas Red-Dextran (70,000 MW, Neutral, Thermo Fisher Scientific, D1830) were administered by intracardiac injection and the animal was fixed in a custom-made stage using the head-attached titanium ring. Images were acquired in a TriM Scope II multi photon system (LaVision BioTec) fitted with a Chameleon Ultra II Ti:Sapphire laser (Coherent) a Chamaeleon Compact optimal parametric oscillator (OPO, Coherent) and a water dipping 16× CFI-75 NA 0.8 objective (Nikon) with a working distance of 2.0 mm. Images were acquired and processed with ImSpector Pro 5.1.347 (LaVision BioTec).

In vivo imaging of blood flow dynamics at cellular resolution in brain blood vessels and calculation of blood flow velocities was based on centreline scans[60]. To ensure flow velocities remained constant, selected vessels were measured twice, at the beginning and end of an experiment. Overview images were captured at the beginning of each imaging session.

**Transmission electron microscopy**. For electron microscopy, 10 ml of PBS and 40 ml of 2% PFA and 2% glutaraldehyde in 0.1 M cacodylate buffer (pH 7.2) were perfused through the left ventricle of an anaesthetized mouse. The brain was removed and further fixed by immersion for 3 h at room temperature (RT). An acrylic brain matrix for mouse (World Precision Instruments, RBMA-200C) and blades were used to obtain coronal slices, which were post-fixed in reduced 1% osmium tetroxide containing 1.5% potassium hexacyanoferrate. The tissue was dehydrated and embedded in epon and 60-nm ultrathin sections were cut on an ultramicrotome (UC6, Leica). The sections were counterstained with uranyl and lead and subsequently imaged on an electron microscope (Tecnai 12 Biotwin TEM, FEI). Representative pictures were documented in imaging plates (Ditabis, Pforzheim).

**Haemorrhagic index calculation**. Quantitation of haemorrhagic index was based on assessment of haemoglobin content in brain tissues. In brief, P10 pups were anaesthetized and perfused with 10 ml of PBS. Brain hemispheres without olfactory bulb and cerebellum were collected in a weighted tube. After addition of 300 μL of double distilled water, the samples were homogenized with a tissue grinder (Pellet Pestle Cordless Motor, Kimble Chase) and centrifuged at 20,000$g$ (30 min, 4 °C).

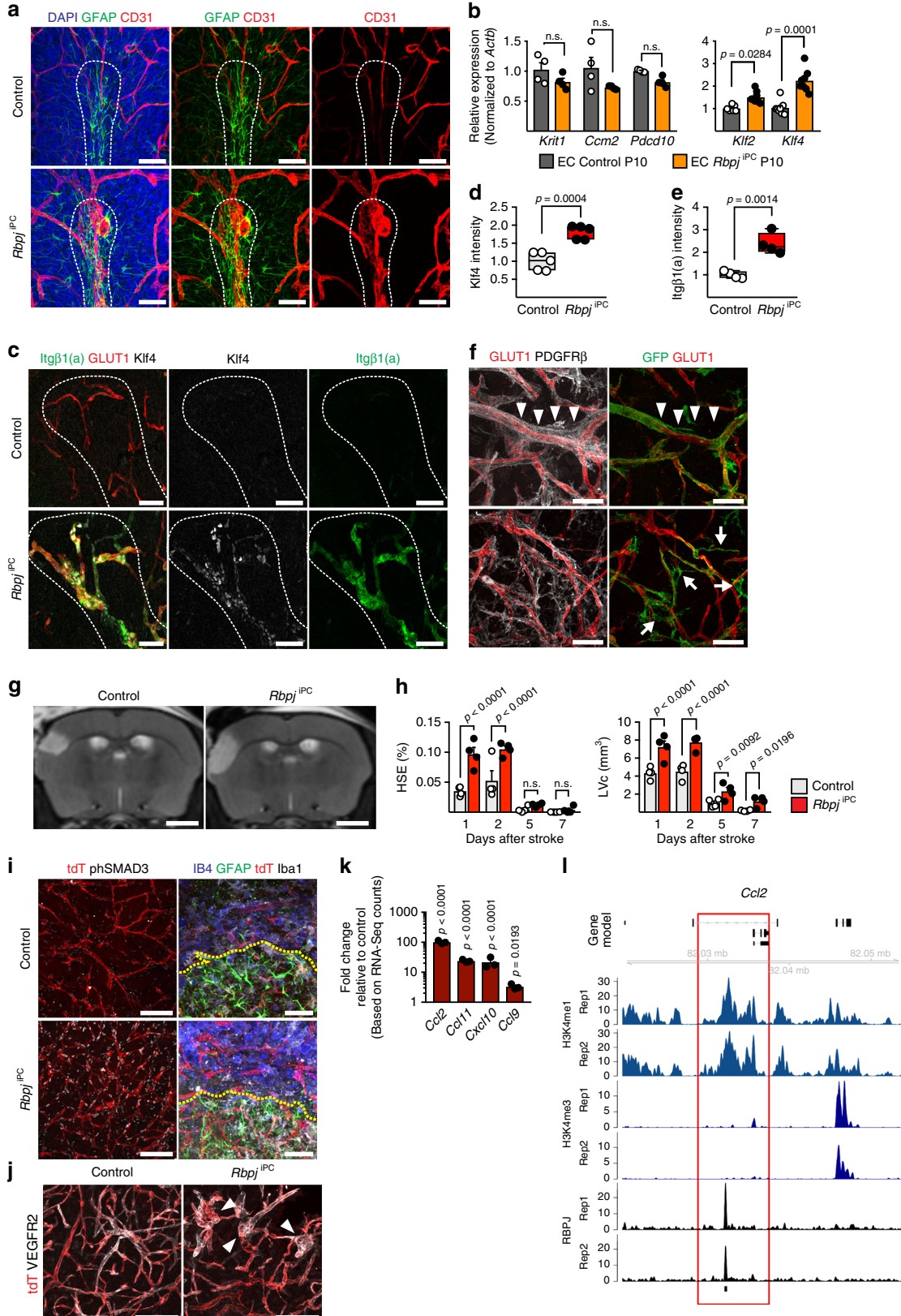

200 μL of tissue homogenate were collected, mixed with 800 μL of Drabkin's reagent (Sigma, D5941) and incubated for 15 min at RT. Absorbance at 550 nm was measured (BioPhotometer plus, Eppendorf) and compared to a standard curve of known haemoglobin concentration.

**BBB analysis and identification of AV malformations**. For BBB analysis, adult animals were injected i.p. with 2% Evans Blue (960 Da, 4 ml kg⁻¹ of body weight,

Sigma, E2129) and 10% fluorescein sodium salt (376 Da, 0.6 ml kg⁻¹ of body weight, Sigma, F6377) in normal saline. The stain was allowed to circulate for 24 h. Afterwards, the animals were transcardially perfused with 20 ml of PBS and 15 ml of ice-cold 4% PFA. Brains were collected and postfixed in 4% PFA ON at 4 °C. Vibratome sections were obtained and imaged by confocal microscopy as already described. Note that sodium fluorescein is used as a reporter of barrier leakage to small molecules (smaller than 0.4 kDa), while Evans Blue reflects albumin (66.5 kDa) extravasation.

**Fig. 7** Molecular similarities with CCMs and impact of *Rbpj* deletion in stroke. **a** Confocal images of cerebellar white matter (dashed outline) in P10 brains. Note dilated vessels (CD31[+], red) and gliosis (GFAP[+], green) in mutants. Scale bar, 50 μm. **b** RT-qPCR analysis in P10 sorted brain ECs. *p*-values, Brown–Forsythe and Welch one-way ANOVA with Tamhane's T2 test (*Krit1*, *Ccm2*, and *Pdcd10*), and Kruskal–Wallis with Dunn's test (*Klf2* and *Kl4*), *n* = 4. **c** Confocal images showing Klf4 (white) and active integrin-β1 (Itgβ1(a), green) in blood vessels (GLUT1[+], red) of P10 *Rbpj*[iPC] cerebellar white matter (dashed outline). Scale bar, 50 μm. **d**, **e** Klf4 (**d**) and active integrin-β1 (**e**) immunosignal fold change intensity in P10 vasculature. *p*-values, Student's *t*-test, *n* = 5. **f** Confocal images showing perivascular PDGFRβ (white) expression after stroke. Note abundance of PDGFRβ[+] GFP[−] cells (arrowheads, top panel) and morphological changes in GFP[+] pericytes (arrows, bottom panel). Scale bar, 50 μm. **g** Representative T2-weight magnetic resonance imaging (MRI) 1 day after stroke. Scale bar, 2 mm. **h** MRI-based quantitation of space-occupying effect attributable to oedema (% HSE) and the lesion volume corrected (LVc, mm³) after stroke. FDR-adjusted *p*-values, one-way ANOVA with two stage linear step-up procedure of Benjamini, Krieger, and Yekutieli multiple comparison correction, *n* = 4. **i** Confocal images showing increased phSMAD3 (white; left panel) and infiltration of inflammatory cells (IB4[+], blue; Iba1[+], white; right panel) in *Rbpj*[iPC] brains after stroke. GFAP staining (green, right panel) indicates boundary between infarct zone and penumbra (dashed line). Vessels are labelled by tdT (red). Scale bar, 100 μm (left) and 50 μm (right). **j** Confocal images showing blood vessel (tdT, red; VEGFR2, white) defects (arrowheads) in *Rbpj*[iPC] mice 7 days post-stroke. Scale bar, 100 μm. **k** Gene expression fold change of pro-inflammatory chemokines in P10 *Rbpj*[iPC] mural cells relative to controls (set as 1). *p*-adjusted values. *n* = 3. **l** RBPJ binding and H3K4me, H3K4me3 distribution at *Ccl2* locus in pericytes. Black bar below RBPJ profiles indicates the position of peak intervals. Two replicates (Rep1 and Rep2) are shown. Error bars represent s.e.m. Source data are provided as a Source Data file

For identification of arteriovenous (AV) malformations, P10 pups were anaesthetized and euthanized by transcardial perfusion of PBS through the left ventricle. Immediately after and without removing the needle used for perfusion, 100 μL of 1% Evans Blue in normal saline were injected followed by 300 μL of latex (Naturlatex, Laguna). Efficient labelling was monitored by visual inspection of main superficial arteries in the sternum and neck region. Brains were collected, fixed in 4% PFA ON at 4 °C, washed in PBS, and imaged in a dissection microscope.

**FACS of brain mural and endothelial cells**. Brain cortices from P5, P7, or P10 pups were collected in ice-cold dissection media (DMEM (Sigma, D6546) supplemented with penicillin/streptomycin (PAA, P11-010) and 25 mM HEPES (Sigma, H3537)) and transferred to digestion media; 37 °C pre-warmed dissection media supplemented with 100 μg ml⁻¹ Liberase DH (Roche, 5401054) and 50 μg ml⁻¹ DNAseI (Sigma, DN25). Samples were minced and digested for 15 min at 37 °C with occasional pipetting to disaggregate tissue pieces. FACS buffer (PBS with 2% FCS) was added and the tissue homogenate was mixed with 1.7 volumes of 22% albumin fraction V (BSA, Carl Roth, 8076.2). After centrifugation (1000*g*, 12 min at RT) the supernatant was removed and the cell pellet resuspended in 5 ml of FACS buffer, filtered through a 40 μm nylon mesh (Corning, 352340) and centrifuged (300*g*, 5 min at RT). The cell pellet, consisting mostly of single cells was resuspended in a suitable volume of FACS buffer together with the following antibodies: rat anti-TER-119-Pacific Blue (1:100, BioLegend, 116232), rat anti-CD45-Pacific Blue (1:200, BioLegend, 103126), rat anti-CD31-FITC (1:100, Invitrogen, RM5201), rat anti-CD13-phycoerythrin (PE) (1:100, BD Pharmingen, 558745), rat anti-CD140a-PE/Cy7 (1:100, eBioscience, 25-1401), and rat anti-CD140b-allophycocyanin (APC) (1:50, eBioscience, 17-1401-82). For cell sorting of samples from animals in which pericytes express GFP, the CD31-FITC antibody was replaced by rat anti-CD31-BV711 (1:100, BD Biosciences, 740680). After 30 min incubation on ice, cells were washed with FACS buffer and resuspended in FACS buffer with DAPI (1 μg ml⁻¹) to exclude dead cells. Cells were sorted directly into RLT Plus buffer (Qiagen, 1053393) supplemented with 1% β-mercaptoethanol (Sigma, M6250) using a FACS Aria IIu (BD Biosciences) with a 70 μm nozzle and FACSDiva software (BD Bioscience, Version 6.0).

Gating strategies to discriminate cell populations based on their immunolabelling were defined according to reference experiments using single-stained samples and FMO (fluorescence minus one) controls.

**RNA extraction and RT-qPCR**. Total RNA from freshly sorted or cultured pericytes and ECs was isolated using the RNeasy Plus Micro Kit (Qiagen, 74034) following the manufacturer's instructions. Whole RNA was reverse transcribed and converted to complementary DNA (cDNA) using the iScript cDNA synthesis kit (BioRad, 170-8890). The following FAM-conjugated TaqMan gene expression probes (all from Thermo Fisher Scientific) were used: *Dll4* (Mm00444619_m1), *Esm1* (Mm00469953_m1), *Myc* (Mm00487804_m1), *Pdgfrb* (Mm00435556_m1), *Anpep* (Mm00476227_m1), *Rgs5* (Mm00654112_m1), *Cspg4* (Mm00507257_m1), *Notch3* (Mm01345646_m1), *Acta2* (Mm00725412_s1), *Tagln* (Mm00441661_g1), *Hey1* (Mm00468865_m1), *Bmp2* (Mm01340178_m1), *Bmp4* (Mm00432087_m1), *Nrp1* (Mm00435379_m1), *Ctgf* (Mm01192932_g1), *Thbs1* (Mm01335418_m1), *Krit1* (Mm01316552_m1), *Ccm2* (Mm00524581_m1), *Pdcd10* (Mm00727342_s1), *Klf2* (Mm00454979_g1), *Klf4* (Mm00516104_m1), *Rbpj* (Mm01217627_g1), *Id1* (Mm00775963_g1), *Id3* (Mm00492575_m1), *Tgfbi* (Mm01337605_m1), *Fn1* (Mm01256744_m1), *Tgfb3* (Mm00436960_m1), *Itga8* (Mm01324958_m1), *Adamts8* (Mm00479220_m1), and *Ccl2* (Mm00441242_m1). VIC-conjugated *Actb* (4352341E) TaqMan probe was used to normalize gene expression after housekeeping gene selection using a TaqMan Array Mouse Endogenous Control Plate (Thermo Fisher Scientific, 4426697).

Quantitative PCR (qPCR) reactions were performed on a CFX96 Touch Real-Time PCR Detection System (BioRad) using the Sso Advanced Universal Probes Supermix (BioRad, 1725281). All relative gene expression analyses were performed using the $2^{-\Delta\Delta Ct}$ method in a minimum of 3 animals per group with triplicate reactions for each gene evaluated.

**RiboTag immunoprecipitation, RNA sequencing, and analysis**. For immunoprecipitation and purification of ribosome-associated RNA, mouse brain cortices from P7 or P10 pups were collected and snap frozen in liquid nitrogen. Polysome buffer (50 mM Tris–HCl pH 7.4, 100 mM KCl, 12 mM MgCl₂, cOmplete ULTRA (Roche), 1% NP-40, 1 mM DTT, 200 U ml⁻¹ RNase inhibitor (M0314, New England BioLabs), 100 μg ml⁻¹ Cyclohexamide, 1 mg ml⁻¹ Heparin) was used for tissue homogenization in a dispersing instrument (T25 digital Ultra Turrax, IKA). Tissue homogenates were centrifuged (20,000*g*, 30 min, 4 °C). 2.5% of the supernatant was kept as input and 50% of the remaining volume was used for immunoprecipitation. To this end, the homogenate was incubated for 8 h at 4 °C with anti-hemagglutinin (HA)-tag Magnetic Beads (MBL, M180-11). After incubation, magnetic beads were washed 3 times for 5 min in high salt buffer (50 mM Tris–HCl pH 7.4, 300 mM KCl, 12 mM MgCl₂, 1% NP-40, 1 mM DTT, 100 μg ml⁻¹ Cyclohexamide) and finally resuspended in RLT Plus buffer. Total RNA was isolated using the RNeasy Plus Micro Kit and assessed using a 2100 BioAnalyzer (Agilent). 100 ng of RNA were used for sequencing library preparation with the TruSeq Stranded Total RNA Library Prep Kit (Illumina) following the manufacturer's instructions. Libraries were validated in the 2100 BioAnalyzer and quantified by qPCR and fluorometric analysis (Qubit 2.0, Invitrogen). 75-bp-end sequencing was performed in a MiSeq (Illumina) using the MiSeq Reagent Kit v3 (Illumina). Experiments were performed in triplicate.

RNA-seq data analysis was performed following a standard workflow. Briefly, quality of the sequenced reads was assessed with FastQC (https://www.bioinformatics.babraham.ac.uk/projects/fastqc/) and aligned to the mouse genome assembly (GRCm38.p6 Ensembl release 92) using TopHat2 (version 2.1.1). Gene expression count data was generated using HTSeq-count (version 0.6.1) with the option -m intersection-nonempty. Differential gene expression analysis across samples was performed using DESeq2 package on protein-coding genes. PCA to assess the overall similarity between the samples was performed on transformed read counts. DEGs were selected using an FRD-adjusted *p*-value cut-off <0.05 and an absolute log₂ fold change > 0.5. For model-based clustering analysis, genes with count per million < 1 in all samples were disregarded and the remaining 14,685 genes were further processed. Raw count data from each sample was normalized with the trimmed mean of *M* values method, using the TCC package. Probabilistic model-based clustering of gene expression profiles from all samples combined was performed using the MBClusterSeq package[23] with the expectation maximization algorithm and 6 clusters. Negative binomial model was used to perform the model-based hybrid-hierarchical clustering. Gene symbols were annotated using biomaRt (BioConductor version 3.1). GO analysis was performed using Advaita Bio's iPathwayGuide (ver. 1.4.0, http://www.advaitabio.com/upathwayguide). Heat maps were generated with Heatmapper[61] (http://heatmapper.ca) or the function heatmap.2 from the R package gplots. GSEA on specific gene sets detailed in the "Results" section was performed with GSEA Desktop v3.0 and version 6.1 of the Molecular Signatures Database (http://www.broadinstitute.org/gsea/msigdb, Broad Institute, MIT) using the following collections: hallmark (H), curated (C2), computational (C4), and GO (C5).

For defining brain pericyte- and SMC-enriched gene candidates, scRNA-seq data[24] was used (http://betsholtzlab.org/VascularSingleCells/database.html). Proposed pericyte-specific markers were defined as genes with >10-fold enriched expression in pericytes with respect to arteriolar and arterial SMC and the opposite was set for identification of SMC-specific markers. In both cases the expression cut

off was set as 100 for the cell population of interest. For comparison of lists of genes, the Compare two lists online tool (http://jura.wi.mit.edu/bioc/tools/compare.php, Bioinformatics & Research Computing, Whitehead Institute, MIT) was used.

**Primary brain pericyte isolation and culture**. Brains from 6–8 weeks old mice homozygous for *Rbpj*-floxed (*Rbpj*$^{lox/lox}$), devoid of olfactory bulb and cerebellum, were minced thoroughly before digestion with Papain dissociation system (Worthington, LK003150) for 70 min at 37 °C. The tissue was further disrupted by passing it 10 times through 19- and 21-gauge needles consecutively. The homogenized tissue was mixed with 2.5 volumes of 22% cell-culture grade BSA (Sigma, D8537) and centrifuged (1000*g*, 10 min at RT) to remove the myelin. The cell pellet was washed once with EBM-2 media (Lonza, CC-3156 and CC-4176), centrifuged (300*g*, 5 min at RT) and resuspended in EBM-2 media for seeding in 2% collagen type 1- (BD Biosciences, 354249) coated wells. The next day, cells were washed 3 times with PBS and fresh EBM-2 media was added. When confluence was reached cells were passaged 1:4. Starting from the third passage culture media was changed to pericyte medium (ScienCell, 1201). High-purity pericyte cultures are established around passage 5. Expression of known markers of pericytes was assessed by RT-qPCR and immunostaining. All experiments performed with cultured pericytes were done with cells between passage 12 and 20. All cultured cells were maintained at 37 °C in a humidified atmosphere with 5% $CO_2$.

**In vitro induction of recombination**. For induction of gene recombination in brain primary pericytes from *Rbpj*$^{lox/lox}$ mice, either TAT-Cre or Cre-expressing lentivirus were used. pTAT-Cre (Addgene, 35619) was a gift from Steven Dowdy. TAT-Cre protein was produced in Rosetta 2(DE3)pLysS *Escherichia coli* cells (Novagen, 71403) and purified on Ni-NTA Agarose (Qiagen, 30210) and Endo-Trap HD1/1 (Hyglos, 800053). Lentivirus were produced in HEK293T cells co-transfected with the packaging plasmids psPAX2 (Addgene, 12260), pMD2.G (Addgene, 12259) and either pUltra-Blasticidin or pUltra-Cre-Blasticidin using Fugene6 transfection reagent (Promega, E2691). pUltra (Addgene, 24129) was a gift from Malcolm Moore; the cDNAs encoding Cre and Blasticidin resistance were subcloned into the *Bam*HI, *Nhe*I, and *Sal*I restriction sites.

TAT-Cre, or an equal amount of BSA (negative control), were diluted in MEM media (Gibco, 31095-029), filtered in 0.22 µm low binding protein filter units (Millex-GV, SLGV013SL) and added to pericytes seeded 24 h earlier. 15 h later cells were washed once in PBS and pericyte medium added. Gene recombination analysis or experimental procedures were done 48 h after removal of TAT-Cre. Alternatively, freshly seeded pericytes were incubated with conditioned media from HEK293T cells producing either lenti-Cre or lenti-GFP for 24 h. After infection pericytes were washed 4 times with PBS and left in culture for 48 h before evaluation of gene recombination or experimental analysis.

**Collagen gel contraction assay**. Assessment of in vitro contractility of brain pericytes after *Rbpj* deletion was performed with gel contraction assay[62]. In brief, recombined or control pericytes were suspended at $1 \times 10^5$ cell ml$^{-1}$ in 1.0 mg ml$^{-1}$ type 1 collagen dissolved in 0.8× DMEM supplemented with 8% FBS and with pH adjusted by addition of 0.1 M NaOH. The cell–matrix mixture was added drop-wise in wells of a 24-well plate and was allowed to solidify for 20 min in a cell culture incubator. Pericyte medium was added on top of the solidified matrix to induce free contraction. After 18 or 30 h pictures of the wells were taken with a dissection microscope and the area of the gels was measured manually in Velocity for quantitative evaluation. The contraction index was calculated as (area of the well − area covered by gel) per (area of the well) and normalized to the control (set as 1).

**Chromatin immunoprecipitation (ChIP), library, and sequencing**. ChIP was performed essentially as previously described[63]. The following antibodies were used: H3 (Abcam, ab1791), H3K4me1 (Abcam, ab8895), H3K4me3 (Diagenode, pAb-003-050), and RBPJ (Cell Signaling, #5313). Libraries were prepared using the Diagenode MicroPlex Library Preparation kit v2 (Diagenode) following the manufacturer's instructions with few modifications. Libraries were purified with Agencourt AMPure XP Beads (Beckman Coulter, #A63881), quantified, analyzed on a Tapestation device (Agilent), and pooled. Finally, sequencing was performed by Centro de Análisis Genómico (CNAG-CRG), Spain.

**ChIP-seq data analysis**. Fastq files were controlled for quality issues using fastqc (https://www.bioinformatics.babraham.ac.uk/projects/fastqc/). Read alignment against the GRCm38 mouse reference genome version 92 (downloaded from Ensembl: Mus_musculus.GRC_m38.92.dna.primary_assembly.fa) was performed using bowtie version 1.1.2 with parameters *-k 1 -m 3*. The corresponding index was built using bowtie's bowtie-build function with the above-mentioned sequence as input. Similarly, we used gene annotations from Ensembl (Mus_musculus. GRC_m38.92.gtf). Duplicate removal was performed using Picard's MarkDuplicates and Samtools rmdup function. Coverage vectors were generated with Deeptools bamCoverage function using FPKM normalization. Visualization of binding profiles was done using the R/BioConductor package Gviz. RBPJ peak calling was done using Macs2[64] and Peakranger[65]. Peaks within 2 kb were

subsequently merged together. For robust detection of peaks, we used the intersection between both methods and both replicates asking for a minimum of 3 overlapping peak calls using both methods on two replicates. The resulting set was filtered against blacklisted chromatin regions as detected by ENCODE. In order to identify overrepresented functional terms amongst RBPJ binding site associated genes we used GREAT[66] using standard settings. For heatmap representation of binding data across RBPJ sites, coverage vectors were collected in defined intervals across RBPJ binding peaks using Deeptools computeMatrix and plotted via Deeptools PlotHeatmap function using *k*-means clustering parameter set to 5. In order to compare different experimental samples amongst each other we calculated the Manhattan distance across using genome-wide read counts collected in 1 kb bins. The resulting matrix was then plotted after hierarchical clustering (using Euclidean distance and complete linkage method with R's hclust function).

For motif identification, we determined the site of maximum coverage within each peak and selected the corresponding DNA sequence from −50 bp to +50 bp around that maximum as input for MEME-ChIP[67] as well as for rGADEM. All downstream analysis was done within R. Manipulation of sequencing reads was done using Rsamtools and genomic intervals were represented as GenomicRanges objects[68]. RBPJ associated genes and enrichment of corresponding functional terms and annotations were identified with GREAT using the gene association rule 1 with default settings. All downstream processing and visualization were done in R. Analysis routines are available upon request.

**Distal middle cerebral artery occlusion (dMCAO)**. The dMCAO procedure were performed as previously described[69] in >20-week-old (adult) male mice. Surgical procedures started 1 week after the last tamoxifen administration. The animals were anaesthetized by intraperitoneal injection of ketamine (80 mg kg$^{-1}$) and xylazine (16 mg kg$^{-1}$). The skin incision was performed to gain access to the temporal muscle, which was partially removed using electrocoagulation forceps with the high-frequency generator set at 5–7 W. After generating the muscle flap, the middle cerebral artery (MCA) was identified and the skull above the MCA branch was thinned using a microdrill. After removing the bone, the MCA was coagulated proximally and distally to the vessel bifurcation using bipolar forceps with the energy of the high-frequency generator set to 3–5 W. After ensuring that the blood flow is permanently interrupted, the temporal muscle was fixed at the muscle insertion to cover the bone defect and the wound was sutured. After all surgical interventions, animals received subcutaneous volume substitution with saline, analgesic treatment (paracetamol 2 mg ml$^{-1}$ in drinking water), and were monitored as established in the approved protocols.

**Magnetic resonance imaging (MRI)**. Animals were subjected to MRI on days 1, 2, 5, and 7 after dMCAO operation. MRI was performed using a 7 Tesla rodent scanner (Pharmascan 70/16 US, Bruker Biospin) with a 16-cm horizontal bore magnet and a 9 cm (inner diameter) shielded gradient with a H-resonance-frequency of 300 MHz and a maximum gradient strength of 300 mT m$^{-1}$. For imaging a $^1$H-RF quadrature-volume resonator with an inner diameter of 20 mm was used. Data acquisition and image processing were carried out with Paravision 4.0 software (Bruker). During the examination mice were anaesthetized with isoflurane (induction with 2% isoflurane and maintenance with 1.5% isoflurane in 70% $N_2O$ and 30% $O_2$ via a vaporizer) and placed on a heated circulating water blanket to ensure constant body temperature of 37 °C. Brains were imaged with a T2-weighted 2D turbo spin-echo sequence with imaging parameters TR per TE = 4200 per 36 ms, rare factor 8 and 4 averages. In each case, 32 axial slices with a slice thickness of 0.5 mm, a field of view of 2.56 × 2.56 cm and a matrix of 256 × 256 were positioned over the brain from the olfactory bulb to the cerebellum.

Lesion volume quantitation was performed with Analyze 10.0 software (AnalyzeDirect, Inc.) and OsiriX DICOM Viewer (Pixmeo) using T2-weighted images where hyperintense ischaemic areas are defined with a region of interest tool. This procedure enables a threshold-based segmentation and results in a 3D object map of the whole stroke region. The whole object map total volume was automatically calculated. The space-occupying effect due to brain oedema, expressed as the volume increase of the affected hemisphere (%HSE), and the oedema-corrected lesion volume (LVc) were calculated according to standardized protocols[70].

**Statistical analysis**. All data are presented as mean ± standard error of the mean (s.e.m.). No statistical methods were used to predetermine sample size; instead sample number was defined according to previous experience and reproducibility of the results across several independent experiments. For all experiments done the number of animals (*n*) analyzed per group was derived from at least two different litters. No animals were excluded from the analysis and no randomization or blinding was used. Unless otherwise indicated, results are based on three or more independent experiments to guarantee reproducibility of the findings.

Sample distribution was assessed with the D'Agostino & Pearson omnibus test or the Shapiro–Wilk normality test depending on sample size. Unpaired two-tailed Student's *t*-test was used to determine statistical significance when comparing two independent groups with normal distribution and no significant difference in their standard deviation (SD); alternatively, Welch's *t*-test was used for normal distributed data with unequal variances. When comparing two independent groups

where at least one of them did not fit the normality criteria, unpaired two-tailed Mann–Whitney test was used. For analysis of statistical significance in comparisons involving more than two groups with normal distribution, ordinary one-way ANOVA with Tukey's (when comparing the mean of each group with the mean of every other group), Sidak's (for comparing the means of preselected pairs of groups) or Dunnett's (when comparing the mean of each group with the mean of a control group) post-hoc test was used. For comparison of more than 2 groups with normal distribution but unequal standard deviations, Brown–Forsythe and Welch one-way ANOVA tests were performed with Tamhane's T2 post-hoc test. For multiple comparisons in which at least one group does not comply with normal distributed data, Kruskal–Wallis and Dunn's post-hoc tests were used. In all cases statistical significance was assessed with a 95% confidence interval, therefore $p$ value <0.05 was considered significant. For the analysis of stroke lesions volume along time, ordinary one-way ANOVA with multiple comparisons for selected pairs (control and $Rbpj^{iPC}$ for each time point of analysis) was used; correction for multiple comparisons was performed by controlling the FDR using the two-stage step-up method of Benjamini, Krieger, and Yekutieli with Q value of 0.05. Statistical analysis was performed using GraphPad Prism 7 (GraphPad, version 7.0b) or the R statistical environment (http://r-project.org).

Whenever boxplots are used, the centre line coincides with the second quartile (the median), the bounds of the box correspond to the first and third quartile, while the end of the whiskers represent the minimum and maximum of all the data.

**Reporting summary**. Further information on research design is available in the Nature Research Reporting Summary linked to this article.

## Data availability

All relevant data supporting the results of the present study are included within the article, its Supplementary Information files, and can be obtained from the corresponding author on reasonable request. The original RNA-seq and ChIP-seq data are available from the Gene Expression Omnibus (GEO) database under accession codes GSE117083 and GSE120482, respectively. An UCSC browser session for peak visualization has been created under the following link: [https://genome-euro.ucsc.edu/s/MarekB/mm10_RBPJ_NatCom]. A Source Data file is included which contains the raw data underlying the reported averages in all figures and supplementary figures. A reporting summary for this Article is available as a Supplementary Information file.

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

## Acknowledgements

Funding was provided by the Max Planck Society, the University of Münster, the Deutsche Forschungsgemeinschaft (DFG FOR 2325 and Cells-in-Motion Cluster of Excellence) and the TNE Program of the Fondation Leducq. R.D.-H. was partially funded by the European Union (Marie Curie ITN VESSEL). B.D.G. is supported by a Research Grant of the University Medical Center Giessen and Marburg (UKGM) and M.B. by the collaborative research grant TRR81-Z1. In addition, work performed in T.B. laboratory was supported by the collaborative research grant TRR81-A12 and BO1639/9-1 (DFG), the Excellence Cluster for Cardio Pulmonary System (ECCPS), and the Behring-Röntgen foundation (BR04). The authors thank S. Volkery and M. Stehling for experts advise with confocal microscopy and flow cytometry, respectively; D. Zeuschner and K. Mildner for electron microscopy; Charité Core Facility 7T (Charité University Medicine Berlin) for experimental MRIs; M. Vanlandewijck and K. Nahar for help with the *Pdgfb*^Ret/Ret^ analysis; S. Dowdy for TAT-Cre plasmid and M. Moore for pUltra plasmid; G. de Luxán for critical reading of the manuscript; H.W. Jeong for help with RNA-seq experiments; and Centro de Análisis Genómico (CNAG-CRG), Spain, for sequencing the ChIP samples.

## Author contributions

R.D.-H. and R.H.A. designed the study, interpreted the results, and wrote the manuscript. K.K. analyzed RNA-seq data, optimized protocols for immunoprecipitation, generated lentivirus constructs, and purified TAT-Cre protein. R.D.-H. directed H.A. and together carried out the immunohistochemistry, qPCR, confocal imaging, cell culture, and quantifications. M.N.-K. and P.V. did the dMCAO stroke surgeries, T2 MRI imaging and quantitation, collected stroke samples, and helped in the interpretation of results. B.D.G., F.F., and T.B. performed ChIP experiments, prepared libraries, and interpreted the results with the help of M.B. and T.Z. who performed the bioinformatic analysis on the ChIP-seq data. M.G.B. performed blood flow measurements using two-photon microscopy. H.M.E. helped in the initial characterization of the *Pdgfrb-CreERT2* and establishing mutant colonies. S.A. generated *Pdgfrb-CreERT2* mice.

## Additional information

**Competing interests:** The authors declare no competing interests.

