## [Peer Review File · Nature Communications]

Reviewers' comments:

Reviewer #1 (Remarks to the Author):

Here, Diéguez-Hurtado et al. utilize *Pdgfrb-CreER;Rbpjlox/lox* mice to postnatally delete *Rbpj* gene, encoding the transcription factor RBP-J κ , from mural cells. Complete deletion of *Rbpj* from mural cells caused an alteration in pericyte gene expression, increased TGF β signaling, and vascular phenotypic changes, including severe hemorrhages, reduced capillary density, twisted and tangled capillaries, enlargement of veins, and vascular malformations. Most arteries and arterioles remained unchanged, and pericyte coverage was unaltered. These phenotypic changes were not observed with mural cell deletion *Pdgfrb-CreER;iDTR*, or mural cell depletion *PdgfrbRet/Ret* mutant mice implicating that the *Rbpj*-related are independent of pericyte loss. Also, the authors show that adult *Pdgfrb-CreER;Rbpjlox/lox* mice have a larger infarct upon distal middle cerebral artery occlusion. Altogether, these results are quite intriguing, but several major concerns exist that dampen the overall enthusiasm for this manuscript as described below:

Major comments:

1. *Pdgfrb-CreER* mice are not pericyte specific and the authors should edit their language to indicate conditional deletion in “mural cells”.
2. *Pdgfr β* is expressed by both pericytes and vascular smooth muscle cells, and some reports also indicate that neurons may express *Pdgfr β* (*J Neurochem.* 2006 Jul;98(2):588–600; *J Neurosci Res.* 2010 May 1;88(6):1273–1284). Therefore, any of these cell types could play a causative role in modifying the identified vascular phenotype. This particularly seems to be the case since deletion or depletion of pericytes does not mimic the phenotype of deletion of *Rbpj*. Furthermore, it is clear that there is a change in vascular smooth muscle cell coverage and morphology. This makes the overall data very difficult to interpret.
3. The authors should examine endothelial junctions and pericyte-endothelial junctions. There is a *Rbpj* binding site on the N-cadherin promoter (Li et al. *Dev Cell.* 2011 Mar 15;20(3):291-302), and N-cadherin is abundantly expressed by pericytes and mural cells (Vanlandewijck et al. *Nature.* 2018 Feb 22;554(7693):475-480) and underlies formation of direct peg-and-socket contacts between pericytes and endothelial cells (*Dev Dyn* 218: 472–479, 2000; Zheng et al. *Sci Rep.* 2016 Jul 28;6:30622) to maintain BBB stability. Endothelial junctions are also known to be disrupted in cavernous malformations (*Nature.* 2013 June 27;498(7455):492-6; *Stroke.* 2015 May;46(5):1337-43; *J Neurol*

Neurosurg Psychiatry. 2001;71:188–192; Clin Neurol Neurosurg. 2013 Apr;115(4):438-44; Neurosurgery. 2000;46:1454–1459).

4. Following Rbpj conditional deletion, is the basement membrane enlarged and multilayered, as is characteristic of CCMs?

5. Were the hemorrhaged areas of the vessel poorly covered by mural cells or astrocytes, as is characteristic of CCMs?

6. The authors should better characterize key features of the observed capillary and arteriovenous malformations (including effects of mural cells, endothelial cells, basement membrane, associated astrocyte endfeet, etc.).

7. The authors should better characterize what is happening to endothelial cells. For example, what is causing the hemorrhages? Are endothelial tight junctions forming properly?

8. Cavernous malformations are typically congenital; do the authors observe the same vascular phenotype in mice with congenital conditional Rbpj deletion compared to postnatal deletion at P1-3?

9. The authors should report the approximate number of malformations by regional distribution. The observed phenotype seems to resemble familial CCMs (caused by mutations in CCM1, CCM2, CCM3) which typically develop hundreds of cortical malformations, in contrast to sporadic CCMs that develop fewer, subcortical malformations. Please comment.

10. Rbpj transcription factor is reported to be downstream of Notch signaling in endothelial cells (Li et al. Dev Cell. 2011 Mar 15;20(3):291-302). In addition to mural cells (data shown), do the authors observe disrupted Notch signaling in endothelial cells? Is there altered cellular cross-talk between pericytes/SMCs and endothelial cells? The authors should also conduct RNA-seq of endothelial cells in Pdgfrb-CreER;Rbpjlox/lox mice to answer these questions.

11. The authors note that capillaries are dilated; do the authors measure any significant differences in diameter of capillaries?

12. Cavernous malformations are associated with reduced blood flow in these segments of the vasculature. What are the functional physiological consequences of deletion of Rbpj? For example, do *Pdgfrb-CreER;Rbpjlox/lox* mice have altered cerebral blood flow upon stimulus?

13. It is surprising that the authors chose to use animals with hemizygous deletion of Rbpj from *Pdgfrb-CreER* as control mice, and that these mice showed no phenotypic alteration, and were undistinguishable from Cre-negative littermates given the drastic phenotype of *Pdgfrb-CreER;Rbpjlox/lox* mice. Control mice with normal levels of Rbpj should be included in all figures.

14. The increased *Thbs1* signal in Figure 4I appears to be in vessels larger than capillaries.

15. The *phSMAD3* staining in Figure 4b, and the *phSMAD1/5* staining in Figure 4d appear to be nuclei that are too numerous to be pericyte nuclei. Instead they look like endothelial nuclei.

16. The authors RNA-seq analysis uses a cutoff of $\log_2(0.5)$; this is a liberal cutoff that may identify differential gene expression changes that are not real (applies to Fig 2).

17. Numerous differentially expressed genes and pathways are presented. The authors could do a better job presenting this information more clearly/concisely for ease of the reader to follow the story and the authors' working model of how disrupting signaling mechanisms underlie the resulting CCM-like vascular phenotype.

18. The authors should provide supplementary spreadsheets with a full list of differentially expressed genes.

Minor comments:

1. The citation for Hall et al., should be moved just after the statement that, "Pericytes and, in particular, their contractility has implicated in the regulation of cerebral blood flow", rather than being at the end of the sentence.

2. The authors should confirm that the representative images between groups are at the same magnification in Figure 3e and f.

Reviewer #2 (Remarks to the Author):

The manuscript by Diéguez-Hurtado and colleagues utilizes state of the art inducible genetics in mice to study the role of RBPJ, a transcriptional repressor and central effector of Notch signalling, selectively in post-natal pericytes and vascular smooth muscle cells in the brain. The authors carefully confirm specificity of the *Pdgfrb-CreERT2* driver line in the brain, and study the effects on brain vasculature, pericyte coverage, morphology, gene expression, as well as the neuronal environment following inducible deletion of RBPJ at P1,2 and 3. By P10, the authors find dramatic vascular defects in many but not all brain areas, and incipient changes at P7. The authors undertake a highly comprehensive phenotypical characterization, provide beautiful illustrations and a wealth of informative controls. They further investigate transcriptome of the pericytes using the ribotag approach. The authors find that vascular lesions that carry similarities with CCM associate with increased endothelial cell proliferation and pericyte gene expression changes indicating strong upregulation of TGF β and BMP signalling, increased matrix production including matrix components involved in TGF β binding and activation. They further provide convincing evidence for endothelial increases in phospho SMAD1/5 and both pericyte and endothelial increases in phospho SMAD2/3.

Surprisingly genetic inactivation of Notch2 and 3 in pericytes, or general Notch inactivation in pericytes by *dnMaml* expression, as well as activation of Notch using NICD do not phenocopy the RBPJ deletion effects. Neither does pericyte depletion using genetic DTA expression in the same population.

The authors therefore conclude that RBPJ has an important repressor role in postnatal pericytes in the brain, unlikely connected to Notch activity.

Intriguingly, the present work provides evidence for a detrimental effect of pericyte dysregulation as potential driver of vascular lesions and compromising the neurovascular unit in postnatal brains. As such, the work is highly noteworthy, and together with the sophisticated and detailed analysis, the wealth of data, it will be of great interest to both the specialist and general readership.

The question whether or not this is a model that will help us understand potential deleterious effects of pericyte dysregulation in adulthood remains however open. Surprisingly, deleting RBPJ in pericytes in the adult brain does not cause any phenotype. This in itself is a striking observation, as one would expect that silencing of the gene loci that drive the changes in the juvenile brain would need to continue in order to protect vascular homeostasis in the adult. So why does RBPJ deletion in the adult have no effect? Is that because RBPJ is no longer actively transcribed, and there is a pool of RBPJ protein that is sufficient to suppress the loci in the adult quiescent state? Or do we need to suspect further chromatin changes that silence the same loci even in the absence of RBPJ protein? Answering these questions will help to assess whether the reported phenotype merely reflects a

derailed differentiation process of pericytes caused by an artificially induced derepression, or whether the underlying biology might in fact predict important functions that are at play in human disease processes. I realize that answering these questions in detail experimentally may entail many new and different approaches to study chromatin and would go beyond the scope of the present work. In the absence of new data, it would seem however appropriate to discuss the reasons for no overt phenotype in the adult in a little more detail.

The final observation of increased stroke lesions is less surprising, but does indicate that an activation state is a prerequisite for the strong effects of pericytic RBPJ deletion. Here may lie the opportunity to at least comment on whether increased RBPJ protein turnover/expression in pericytes is part of the activation state and may therefore provide a technical explanation for the lack of phenotype in adult.

Minor comments and corrections:

Line 66: word missing "has been implicated"

Line 162: the authors probably mean "sensitive" not "sensible"

Line 206, figure 2g,h Does this indicate partial differentiation of pericytes towards VSM cells? Is it a consequence of detachment? does it mean that VSM usually are lower in RBPj expression or is this because Notch is active in VSM and therefore these loci not repressed?

Line 255, word missing after "post-natal"

Line 261, should read "normally"

Line 328, word missing

Line 330, spelling metalloprotease

Line 381, What happens to endothelial cell proliferation after stroke in the mutants?

Line 398, are there quantitative differences in RBPj expression levels between pericytes and vascular smooth muscle on arteries, veins and capillaries that could explain the differential effects on the different vessel segments?

Line 406, this discussion again raises the question whether RBPj is naturally low in arterial vSMC, possibly explaining the lack of phenotype in arteries in the study.

Reviewer #3 (Remarks to the Author):

This is a comprehensive and rigorous study concerning the role of this transcription factor in pericyte biology. Combining the main data and the extended data provided, this manuscript required considerable time and labor to read through the wealth of figures and data. This is not a criticism, merely a statement of the comprehensive nature of the study.

As stated above, I found the work to be quite comprehensive and rigorous. The rationale for each experiment was very clear and the data was clearly described and illustrated (although toggling back and forth between the main data and the extended data was quite confusing). I have no scientific concerns regarding the work.

My only concern is that if these vascular lesions truly resemble cavernous malformations, it is surprising that no one has reported this gene to be mutated in any families or individual cases. Furthermore, the Discussion section does not discuss how this work relates to CCM, other than that the lesions resemble CCMs. How does this inform us of CCM pathobiology and what does it mean that a TF in pericytes can cause the same vascular lesions (with some of the same molecular signatures) as loss of CCM proteins in the endothelial cell compartment. If the case is to be made that these lesions resemble CCMs, then it would be important to discuss this discovery in the context of the rich literature on CCM pathobiology.

First of all, we would like to sincerely thank the reviewers for their time and thoughtful comments, which have enabled a substantial improvement of our manuscript. While we will provide detailed point-by-point responses further below, we wish to start with a summary of the most important changes and additions that you will find in this revised version.

As requested by the reviewers, we have carefully compared the brain vasculature of young pups that lack the Cre transgene (*Pdgfrb-CreERT2^{+/+}*) to that of mice with single allele deletion of *Rbpj* (*Rbpj^{+iPC}*). This ruled out the emergence of detectable phenotypic alterations in the latter (Supplementary Fig. 2). Moreover, we have confirmed that the cerebrovascular alterations detected in mural cell-specific *Rbpj* mutants are not recapitulated after gene deletion using the astroglial Cre-driver *GFAP-CreERT2* (Revision Fig. 1). These new results are in accordance with the conclusions of the original manuscript and further reinforce the mural cell-specific importance of *Rbpj* in the cerebral vasculature.

Furthermore, we have refined the characterization of the vascular phenotype by including quantitative analysis of capillary diameters (Fig. 1g) and performing a more detailed assessment of astrocyte endfeet polarization (Fig. 2c, d and Revision Fig. 3), endothelial tight junctions' integrity (Supplementary Fig. 3f, g), and EC apoptosis (Supplementary Fig. 3i, j). Likewise, we have broadened our interrogation of gene expression changes in endothelial cells with special attention to molecules involved in endothelial-mural cell adhesion and crosstalk (Revision Fig. 2) as well as to the activation of angiogenesis-related signalling cascades, such as the Notch pathway (Revision Fig. 4). Altogether, the new data provide a more comprehensive understanding of the profound alterations elicited by *Rbpj* deletion in brain mural cells and provide deeper insight into the defective communication between mutant pericytes and endothelial cells.

In order to expand the physiologic relevance and functional implications of the findings described so far, we have performed an *in vivo* blood flow velocity assessment in P10 control and mutant mice by transcranial two-photon microscopy after intravenous injection of a fluorescent dye. These results show that upon *Rbpj*-deletion blood flow speed remains within normal parameters in arteries but is significantly reduced in veins, reflecting the drastic changes in vessel diameter and in the morphology of mutant capillaries and veins (Fig. 2e, f).

With the purpose of gaining further insight into the molecular mechanisms responsible for the altered transcriptional signature and the pathogenic conversion of pericytes, we have analysed RBPJ binding sites in cultured pericytes by ChIP-Seq (Fig. 6 and Supplementary Fig. 12). Noteworthy, RBPJ was found to bind 122 genes that are upregulated in *Rbpj* mutants *in vivo* including critical regulators of TGF β signalling and neuroinflammation such as *Thbs1*, *Tgfb3* and *Ccl2*. These results point out that RBPJ binds transcriptional domains of relevant genes identified in other parts of the data. These new results provide additional evidence for a direct role of RBPJ in the pericyte-mediated control of TGF β activation within the neurovascular unit.

Additionally, in order to better explain the distinct response to *Rbpj* deletion in different mural cell subsets and the phenotypic differences in early postnatal or adult stages, we have extended our analysis of Notch activation profiles in mural cells by detailed examination of two different reporter mouse lines, namely *Hey1-EGFP* and *CBF:H2B-Venus* (Supplementary Fig. 7). These new results allow us to conclude that Notch signalling is preferentially activated in arterial vascular smooth muscle cells but remains inactive in pericytes during the first two weeks of postnatal development. In older animals, RBPJ-

mediated transcriptional activation in pericytes becomes more frequent suggesting a phenotypic switch related to maturation. Altogether, this new line of evidence correlates well with the proposed role of RBPJ as a repressor in early postnatal pericytes, a behaviour that is recapitulated after injury.

Moreover, while preparing the source data file, we have further corrected statistical approaches when necessary and have strengthened the quantitative analysis of gene expression for critical molecules involved in phenotypic conversion (i.e. *Thbs1*, *Tgfb3*, *Acta2*, *Tagln*). Likewise, we have improved the quality and representativeness of relevant immunostaining experiments (i.e. *Thbs1*) following reviewer's advice and provide now supplementary tables for differentially expressed genes and the results from the ChIP-Seq analysis.

We believe that the new experiments performed, together with the extensive revision of the text and reorganization of the figures, have successfully addressed all critical concerns and allowed us to substantially improved the quality and relevance of our study.

Point-by-point response: the full referee comments (black) are followed by our answers and comments (blue).

Reviewers' comments:

Reviewer #1 (Remarks to the Author):

Here, Diéguez-Hurtado et al. utilize *Pdgfrb-CreER;Rbpjlox/lox* mice to postnatally delete *Rbpj* gene, encoding the transcription factor RBP-J κ , from mural cells. Complete deletion of *Rbpj* from mural cells caused an alteration in pericyte gene expression, increased TGF β signaling, and vascular phenotypic changes, including severe hemorrhages, reduced capillary density, twisted and tangled capillaries, enlargement of veins, and vascular malformations. Most arteries and arterioles remained unchanged, and pericyte coverage was unaltered. These phenotypic changes were not observed with mural cell deletion *Pdgfrb-CreER;iDTR*, or mural cell depletion *PdgfrbRet/Ret* mutant mice implicating that the *Rbpj*-related are independent of pericyte loss. Also, the authors show that adult *Pdgfrb-CreER;Rbpjlox/lox* mice have a larger infarct upon distal middle cerebral artery occlusion. Altogether, these results are quite intriguing, but several major concerns exist that dampen the overall enthusiasm for this manuscript as described below:

Reply: We appreciate the critical comments and we hope that the extensive revision detailed below will address all concerns raised and thereby satisfy the reviewer.

Major comments:

1. *Pdgfrb-CreER* mice are not pericyte specific and the authors should edit their language to indicate conditional deletion in "mural cells".

Reply: We agree with this reviewer in that *Pdgfrb* expression is not restricted to pericytes but shared by other mural cells, which would therefore be susceptible to conditional deletion using our *Pdgfrb(BAC)CreERT2* mouse line. Nevertheless, it should be pointed out that *Pdgfrb* expression among mural cells is higher in pericytes with respect to artery-associated vascular smooth muscle cells (SMCs), as was recently shown by single-cell sequencing¹.

Moreover, when it comes to description of phenotypic alterations, we believe that referring to pericytes is actually more accurate since the most important alterations are restricted to mural cells of capillaries (thus, pericytes) and postcapillary venules and veins, which have been shown to be molecularly alike and different to arterial SMCs. Altogether, we have edited the manuscript to reflect that conditional gene deletion will target all mural cells, but we continue to refer to pericytes when we describe alterations in the mural cells of capillaries and veins.

2. *Pdgfr β* is expressed by both pericytes and vascular smooth muscle cells, and some reports also indicate that neurons may express *Pdgfr β* (J Neurochem. 2006 Jul;98(2):588–600; J Neurosci Res. 2010 May 1;88(6):1273–1284). Therefore, any of these cell types could play a causative role in modifying the identified vascular phenotype. This particularly seems to be the case since deletion or depletion of pericytes does not mimic the phenotype of deletion of *Rbpj*. Furthermore, it is clear that there is a change in vascular smooth muscle cell coverage and morphology. This makes the overall data very difficult to interpret.

Reply: There are controversial reports regarding *Pdgfrb* expression in non-mural cells. Some authors propose that PDGFR β is exclusively expressed in pericytes², while more recent evidence suggests that this molecule is also present in astrocytes and brain fibroblasts^{1,3}. In our hands, the *Pdgfrb(BAC)CreERT2* driven-recombination in the brain cortex is mostly restricted to mural cells (Supplementary Fig. 1a, e, d) although recombination in astroglial cells was occasionally detected, as well as in progenitors of the subventricular zone (SVZ). Likewise, a recent report on the generation of a *Pdgfrb-P2A-CreERT2* mouse model⁴ also mentions scattered recombination in non-mural cells of the brain without further specification of cellular identity.

Since numerous authors have previously addressed the effects of *Rbpj* deletion in different neuronal populations without describing the appearance of vascular defects^{5,6} we focused on ruling out whether inactivation of this gene in astroglia and SVZ could have deleterious effects on blood vessels. In order to do so, we bred *GFAP-CreERT2* transgenic mice⁷ into the *Rbpj*-floxed background and analysed integrity of the cerebrovasculature in P10 mice after tamoxifen administration from P1-P3 in the same manner as done for the mural cell-specific *Rbpj* conditional knockouts (Revision Fig. 1a). Noteworthy, no vascular lesions were detected in such mice (Revision Fig. 1b) despite efficient recombination (Revision Fig. 1c) in the target cell population. Moreover, mural cell coverage and astrocyte-end feet polarization remained unchanged (Revision Fig. 1d) and there were no signs of overt activation of integrin β 1 or overexpression of *Klf4* (Revision Fig. 1e), two important landmarks of pathogenic alteration in the vasculature of *Rbpj*-knockout pericytes. These results show that the identified vascular phenotype is caused mostly, if not exclusively, by the lack of *Rbpj* in mural cells without relevant contribution from recombination elsewhere.

Furthermore, it should be noted that acute pericyte-depletion using the *Rosa26-DTA* model targets the same cellular populations as does conditional deletion of *Rbpj*, since both rely on Cre-mediated recombination. Finally, the particular contribution of pericytes or arterial smooth muscle cells to the vascular phenotype is self-explanatory in the sense that major disruptions of integrity as well as drastic changes in endothelial proliferation and sprouting

Revision Figure 1. Deletion of *Rbpj* in astroglial cells does not compromise vascular integrity.

a, Representation of the *GFAP-CreERT2* transgene and of Cre-mediated recombination of the *Rbpj*^{lox} allele to generate *Rbpj*^{iAC} mutants.

b, Macroscopic images of whole brains and coronal sections from control and *Rbpj*^{iAC} mice at P10 following tamoxifen administration from P1 to P3.

c, Confocal images showing recombination in astroglial cells (identified by GFP, green, expression, yellow arrowheads) in the tdT⁺ (red) brain vasculature of P10 mice without obvious deleterious effects on pericyte (PDGFRβ⁺, white) coverage. Scale bar, 100 μm.

d, Confocal images showing normal astrocyte endfeet (Aqp4⁺, green) investment of brain blood vessels (IB4⁺, PDGFRβ⁺, white) in P10 pups. Scale bar, 50 μm.

e, Confocal images showing unaltered levels of integrin β1 activation (Itgb1a, green) and Klf4 expression (white) in the brain endothelium (GLUT1⁺, red) of *Rbpj*^{iAC} P10 pups with respect to littermate controls. Scale bar, 50 μm.

are limited to the capillaries and veins without affecting arteries. We propose that the different outcome upon *Rbpj*-deletion in pericytes vs. arterial smooth muscle cells is based on the distinct levels of Notch signalling activation in these cell types as it is detailed further below in our reply to reviewer #2.

3. The authors should examine endothelial junctions and pericyte-endothelial junctions. There is a *Rbpj* binding site on the N-cadherin promoter (Li et al. *Dev Cell*. 2011 Mar 15;20(3):291-302), and N-cadherin is abundantly expressed by pericytes and mural cells (Vanlandewijck et al. *Nature*. 2018 Feb 22;554(7693):475-480) and underlies formation of direct peg-and-socket contacts between pericytes and endothelial cells (*Dev Dyn* 218: 472–479, 2000; Zheng et al. *Sci Rep*. 2016 Jul 28;6:30622) to maintain BBB stability. Endothelial junctions are also known to be disrupted in cavernous malformations (*Nature*. 2013 June 27;498(7455):492-6; *Stroke*. 2015 May;46(5):1337-43; *J Neurol Neurosurg Psychiatry*. 2001;71:188–192; *Clin Neurol Neurosurg*. 2013 Apr;115(4):438-44; *Neurosurgery*.

Reply: This is indeed a valuable suggestion and we are grateful to the reviewer for bringing this up. Already in the original submission we provided data showing conserved expression of junctional proteins in the *Rbpj* mutant endothelium and ultrastructural integrity of tight junctions, as shown by transmission electron microscopy (Fig. 2a, b). We have performed additional immunostaining experiments to assess with more detail the expression and localization of the adherens- and tight-junction molecules CDH5, CD31, CLDN5 and ZO1 in the vasculature of mutant mice, with special attention to areas with obvious abnormalities in blood vessel patterning (new data in Supplementary Fig. 3f, g). Despite profound changes in endothelial cell shape and organization, the expression of junction molecules is not substantially compromised and they localize preferentially to intercellular boundaries, which is consistent with the ultrastructural data mentioned above. This suggests that disruption of endothelial junctions is not a primary event although it may contribute to haemorrhagic occurrence as a consequence of vessel rupture and endothelial degeneration.

In cerebral cavernous malformations (CCMs) the existence of substantial intercellular gaps in the endothelium has been thoroughly documented^{8,9}; nevertheless, ultrastructural morphology of endothelial tight junctions lining the caverns appears normal in genetic mouse models of the disease¹⁰.

As requested by the reviewer, we have analysed *Cdh2* (N-cadherin) expression in *Rbpj* mutants by RT-qPCR of freshly sorted brain mural and endothelial cells. N-cadherin is known to mediate adhesion and signalling between endothelial and mural cells¹¹, and thus plays an important role in vascular stabilization¹², which is severely impaired after *Rbpj* deletion. Interestingly, our results point out that *Cdh2* transcripts are significantly downregulated in both cell types as soon as P7 (Revision Fig. 2a, b) suggesting deficient interactions between pericytes and endothelial cells. Evidences of such loose adherence were occasionally detected in *Rbpj*-mutant mice (Fig. 4I in the revised manuscript).

Revision Figure 2. Deficient expression of *Cdh2* in brain mural and endothelial cells of *Rbpj*^{iPC} mutants.

a, RNA-Seq and RT-qPCR expression analysis of *Cdh2* in mural cells from P7 and P10 control and *Rbpj*^{iPC} mouse brain cortex. Error bars, s.e.m., n=3. p-adjusted values.

b, *Cdh2* expression assessment by RT-qPCR in freshly sorted brain ECs from P10 control and *Rbpj*^{iPC} mice. Error bars, s.e.m. p-values, Student's t-test, n=4.

c, Snapshot showing the RBPJ binding along with the distribution of H3K4me1 and H3K4me3 at the *Cdh2* locus in cultured pericytes. Black bars below RBPJ profiles indicate the position of peak intervals. Two replicates (Rep1 and Rep2) of each experiment are shown.

Previous reports have shown RBPJ binding to the promoter region of *Cdh2* in brain endothelial cells and proposed that Notch signalling promotes N-cadherin expression in this cell type¹³. Our own ChIP-Seq data (explained in detail in response to reviewer #2) from cultured mouse primary pericytes also revealed a weak but significant binding of RBPJ at the *Cdh2* locus (Revision Fig. 2c). Nevertheless, it is unclear whether expression of *Cdh2* in brain pericytes of young mice may not be controlled by Notch signalling, as we see very limited Notch reporter activation in these cells (please see also reply to reviewer #2 further below).

4. Following *Rbpj* conditional deletion, is the basement membrane enlarged and multilayered, as is characteristic of CCMs?

Reply: Enlargement of the basement membrane in the brain vasculature of the *Rbpj* mutant animals is an ultrastructural hallmark already mentioned in the original manuscript and further highlighted in the revised version (Fig. 2a, b). Nevertheless, it appears as a continuous and uniform layer, consistently thickened with respect to controls. This, together with the lack of accumulation of amorphous substance in direct contact to the endothelium, are clear differences to what has been reported for CCM lesions. Notwithstanding, the endothelial cells of CCMs and *Rbpj* mutants do share some ultrastructural pathologic features, such as the appearance of enlarged pinocytotic vesicles, intra-cytoplasmic canaliculi and dense protoplasmic protrusions¹⁴.

5. Were the hemorrhaged areas of the vessel poorly covered by mural cells or astrocytes, as is characteristic of CCMs?

Reply: We have shown that overall pericyte coverage is not significantly affected in *Rbpj* mutants (Fig. 2g, h of the revised manuscript). Likewise, astrocyte endfeet coverage of blood vessels does not appear altered in these mice, as indicated by Aqp4 immunosignals around the cerebrovasculature (Revision Fig. 3a). Moreover, in areas of local red blood cell extravasation, both mural cell and astrocytic endfeet coverage are maintained (Revision Fig. 3b and Fig. 2c in the revised manuscript), again suggesting that the primary cause for haemorrhages is not related to defective neurovascular unit composition. This is in line with the absence of brain bleedings in mouse models with acute or chronic mural cell depletion (Supplementary Fig. 4 in the revised manuscript). With respect to CCMs, it should be noted that the poor mural cell and astrocyte endfeet coverage represent secondary changes with a yet uncharacterized relevance in pathogenic progression.

6. The authors should better characterize key features of the observed capillary and arteriovenous malformations (including effects of mural cells, endothelial cells, basement membrane, associated astrocyte endfeet, etc.).

Reply: The revised manuscript contains an extensive characterization of the fundamental aspects behind the vascular abnormalities seen in *Rbpj* mutant mice. We have thoroughly documented changes in mural cells (reduced pericyte identity, increased contractility, altered ECM deposition, changes in adhesion characteristics, alterations in key signalling pathways controlling differentiation), endothelial cells (increased proliferation, decreased sprouting ability, reduced specification of tip cells, altered molecular crosstalk with pericytes,

apoptosis), basement membrane (enlargement and altered molecular composition), associated astrocytes (astrogliosis and swelling despite conserved molecular polarization and vessel coverage by endfeet), and additional effects on the neurovascular unit (focal absence of neuronal markers expression and microglia activation). We believe that our manuscript provides a very comprehensive characterization of relevant aspects governing pericyte-endothelial interactions in the brain vasculature.

Revision Figure 3. Normal astrocyte endfeet coverage in the brain blood vessels of *Rbpj^{fpc}* mutant mice.

a. Confocal images of brain cortex blood vessels (IB4⁺, blue) showing unaltered Aqp4 expression (white) in close proximity to PDGFRβ⁺ mural cells (red) in *Rbpj*-mutant mice or control littermates at P10. GFP expression (green) shows recombined mural cells. Scale bar, 100 μm.

b. High magnification confocal images showing areas of haemorrhaging (red blood cells stained with Ter119, red) in P10 mutant pups. Note that these areas of extravasation are not devoid of recombined pericytes (GFP⁺, green) around blood vessels (tdT⁺, blue) and the unchanged Aqp4 (white) expression and polarization with respect to controls. Scale bar, 25 μm.

7. The authors should better characterize what is happening to endothelial cells. For example, what is causing the hemorrhages? Are endothelial tight junctions forming properly?

Reply: As mentioned in the original manuscript and further highlighted in this revised version (Fig. 2a, b and Supplementary Fig. 3f, g), there are no generalized defects in tight junction molecules expression and localization, nor in the ultrastructural organization of junctions between endothelial cells. We propose that haemorrhaging in *Rbpj* mutant mice first arises as

a consequence of focal endothelial cell degeneration. Hallmarks of such changes in the endothelium, such as accumulation of pinocytotic vesicles and cytoplasmic vacuolization, are obvious already at P7 and worsen as the animals grow older, severely challenging endothelial cell integrity by P10 (Supplementary Fig. 3h in the revised version). Interestingly, similar scenarios of endothelial cell degeneration have been reported in neurovascular-related diseases such as CADASIL¹⁵ and Alzheimer's disease¹⁶. We have furthermore assessed endothelial cell apoptosis by TUNEL assay and immunostaining against active Caspase-3 revealing that in P10 mutant pups show a significant increase in the frequency of apoptotic endothelial cells (Supplementary Fig. 3i, j). Moreover, the additive effects of excessive endothelial cell proliferation, defective sprouting of new branches and increased mural cell contractility are likely to promote vascular instability and vessel rupture in *Rbpj* mutants.

8. Cavernous malformations are typically congenital; do the authors observe the same vascular phenotype in mice with congenital conditional *Rbpj* deletion compared to postnatal deletion at P1-3?

Reply: So far, we have not analysed phenotypic alterations elicited by conditional *Rbpj* deletion during embryonic development. It should be noted that embryonic mesenchymal cells, which give rise to several stromal tissue types in different organs, express *Pdgfrb*¹⁷. Likewise, PDGFR β may have instrumental roles in neuroepithelial and neural crest-derived progenitors¹⁸. Therefore, genetic inactivation using the *Pdgfrb(BAC)CreERT2* transgene during embryonic development will presumably lead to complex phenotypes, which are difficult to interpret given the diversity of targeted cells in several organs.

Familial forms of cavernous malformations are indeed congenital, but they comprise only 6% of all cases^{19,20}. Sporadic lesions, which are more common, are not caused by an inherited genetic mutation but rather by independent, biallelic, somatic mutations²¹.

9. The authors should report the approximate number of malformations by regional distribution. The observed phenotype seems to resemble familial CCMs (caused by mutations in CCM1, CCM2, CCM3) which typically develop hundreds of cortical malformations, in contrast to sporadic CCMs that develop fewer, subcortical malformations. Please comment.

Reply: With the exception of the cerebellum, where we observe focal, CCM-like lesions, the defects affecting veins and capillaries in most other regions of the *Rbpj* mutant brain (i.e. isocortex, caudoputamen, cerebellum) are so widespread that the quantitation of individual malformations is not feasible.

Nevertheless, the distribution of defects in *Rbpj* mutants is similar to the abundance of cortical malformations seen in familial CCMs. Indeed, mulberry-like formations are detected in the cerebellar white matter and along the corpus callosum, beneath the cerebral cortex. Moreover, it should be pointed out that mouse models for CCMs, based on endothelial-specific deletion of any of the genes involved in the familial inherited disease, do not necessarily phenocopy the regional distribution of the lesions seen in human patients. In order to analyse regional specificity and penetrance of the phenotype we have quantified haemorrhage prevalence in different brain regions (Fig.1d).

10. *Rbpj* transcription factor is reported to be downstream of Notch signaling in endothelial cells (Li et al. Dev Cell. 2011 Mar 15;20(3):291-302). In addition to mural cells (data shown), do the authors observe disrupted Notch signaling in endothelial cells? Is there altered cellular cross-talk between pericytes/SMCs and endothelial cells? The authors should also conduct RNA-seq of endothelial cells in *Pdgfrb-CreER;Rbpjlox/lox* mice to answer these questions.

Reply: Besides the already reported downregulation of *Dll4* expression in the endothelium of *Rbpj* mutant mice, we also detected diminished transcription of the Notch target genes *Hes1*, *Hey1*, *Nrarp* and *Notch4* in freshly sorted brain endothelial cells at P10 (Revision Fig. 4a). These results indicate reduced Notch signalling in blood vessels upon *Rbpj* deletion in mural cells. Interestingly, Notch signalling has been shown to be a negative regulator of BMP expression in brain endothelial cells²², and low Notch activity in retinal endothelium is responsible for deregulated angiogenesis associated with excessive proliferation²³. One potential explanation for the reduced Notch activation in *Rbpj* mutants resides in the relative expression of Notch ligands by endothelial and mural cells. Our RNA-Seq data in mural cells

Revision Figure 4. Gene expression analysis of common molecules that govern cellular crosstalk between endothelial and mural cells.

a, RT-qPCR analysis of putative Notch target genes in freshly sorted ECs from P7 and P10 control and *Rbpj*^{iPC} brain cortex. Error bars, s.e.m. p-values, Kruskal-Wallis with Dunn's multiple comparisons test (P7) or one-way ANOVA with Sidak's multiple comparisons test (P10), n=4. n.s., not statistically significant.

b, RT-qPCR analysis of genes mediating endothelial to mural cells crosstalk in sorted brain ECs. Note that besides a discrete decrease in *Tek* (Tie2) expression at P10, all other molecules show normal expression in *Rbpj*^{iPC} mice with respect to controls. Error bars, s.e.m. p-values, one-way ANOVA with Sidak's multiple comparisons test (P7) or Brown-Forsythe and Welch one-way ANOVA with Tamhane's T2 multiple comparisons test (P10), n=4. n.s., not statistically significant.

c, RT-qPCR analysis of genes involved in TGF β signalling in sorted ECs from P7 and P10 control and *Rbpj*^{iPC} brain cortex. Error bars, s.e.m. p-values, one-way ANOVA with Sidak's multiple comparisons test (P7) or Kruskal-Wallis with Dunn's multiple comparisons test (P10), n=4. n.s., not statistically significant.

from control mice show that, among Notch ligands, *Jag1* expression is 5-10-fold higher than any other ligand. Noteworthy, *Jag1* expression in *Rbpj* mutant mural cells is not significantly affected, while there is a trend for overexpression of this ligand in endothelial cells at P10 (Revision Fig. 4b), suggesting that Jagged1-mediated signals remain constant or are even increased. Given the opposing effects of Dll4 and Jagged1 in the endothelium, where the latter antagonizes canonical activation driven by Dll4²⁴, it is feasible to speculate that downregulation of *Dll4* together with unaltered (or increased) Jagged 1 may hamper Notch signalling in brain endothelial cells. Nevertheless, it should be mentioned that these changes arise only in late stages of phenotypic conversion (P10) and are therefore not likely an initial driving force in the development of the mutant phenotype.

As requested by the reviewer, we have carefully checked for changes in the endothelial expression of molecules mediating the crosstalk with mural cells (including *Pdgfb*, *Tek*, *Angpt2* and *Jag1*) as well as in critical components of TGF β signalling (such as *Acvr11*, *Tgfbr1*, *Tgfbr2* and *Eng*, Revision Fig. 4b, c) without finding significant changes. One exception is *Tek* (Tie2), which is known to promote vascular maturation and quiescence²⁵. Altogether these results indicate that pericyte-mediated increase in TGF β and BMP signalling (rather than intrinsic changes in endothelial sensitivity) have a prominent role in the phenotypic alterations described for the brain endothelium. At this stage, we believe that performing RNA-Seq analysis of endothelial cells is beyond the scope of the present manuscript and would drive the attention of the readers to secondary, non-cell autonomous changes that, although interesting and relevant, will not necessarily help to better understand the function of *Rbpj* in mural cells.

11. The authors note that capillaries are dilated; do the authors measure any significant differences in diameter of capillaries?

Reply: Reduced vascular density (i.e. diminished number of branch points) together with enlargement of capillaries are typical characteristic of the *Rbpj* mutant cerebrovasculature which can be appreciated in most of the figures displayed in the manuscript. Similarly, the occurrence of arteriovenous shunts indicates the drastic enlargement of capillary beds, which, in the end, establish abnormal, direct connections between arteries and veins. As requested by the reviewer, we have performed a quantitative assessment of the external diameter of capillaries and found that, in average, mutant mice show a 1.5-fold increase in this parameter (Fig. 1g in the revised version). Although this increase may seem discrete, it should be noted that a small capillary dilation (7%) is able to increase steady state blood flow by 19%²⁶ and that the capillary bed is the largest contributor to hydraulic resistance in the brain²⁷.

12. Cavernous malformations are associated with reduced blood flow in these segments of the vasculature. What are the functional physiological consequences of deletion of *Rbpj*? For example, do *Pdgfrb-CreER;Rbpjlox/lox* mice have altered cerebral blood flow upon stimulus?

Reply: In order to answer this reviewer's question, we have performed *in vivo* measurements of blood flow velocity in brain blood vessels of control and *Rbpj*^{iPC} P10 pups using two-photon microscopy after intravenous injection of Texas Red-labelled dextran. These new results are included in the revised version of the manuscript (Fig. 2e, f). As expected, given that the velocity of flow is inversely related to the total cross-sectional area of the blood

vessels, we could not detect significant changes in the speed of blood flow in arteries when comparing control and *Rbpj*^{iPC} P10 mice. In contrast, blood flow velocity in mutant veins shows roughly a 2-fold dropdown, which we interpret as a consequence of the enlargement in these vessels. Although capillaries in mutant mice are on average enlarged, this effect is not necessarily generalized and local constrictions are a common finding in *Rbpj*^{iPC} mutants.

13. It is surprising that the authors chose to use animals with hemizygous deletion of *Rbpj* from *Pdgfrb*-CreER as control mice, and that these mice showed no phenotypic alteration, and were undistinguishable from Cre-negative littermates given the drastic phenotype of *Pdgfrb*-CreER;*Rbpj*lox/lox mice. Control mice with normal levels of *Rbpj* should be included in all figures.

Reply: Proper interpretation of phenotypic alterations in the mural cells of *Rbpj*-mutant mice demanded in many cases the analysis of control mice where Cre-mediated recombination could drive the expression of fluorescent reporters (i.e. membrane-tagged GFP). We agree with the reviewer in that, the ideal control would have been a littermate with normal *Rbpj* gene dosage (+/+) bearing the Cre transgene in order to drive mural cell-specific activation of the necessary reporter. This approach would normally be possible by mating mice heterozygous for the floxed *Rbpj* allele, with one of the parents (typically the male) carrying the *Pdgfrb*(BAC)*CreERT2* transgene. Such strategy would theoretically yield 50% of pups inheriting one floxed *Rbpj* allele and 25% with either wildtype or floxed homozygous alleles; in all instances, only half of the pups would be expected to bear the Cre-transgene and thus the chance of having conditional knockouts or “ideal controls” in each litter would be reduced to 12.5%. Besides being very animal-costly and largely inefficient, this approach is not feasible in our particular setup due to the integration site of the *Pdgfrb*(BAC)*CreERT2* transgene, which lies in close proximity to the *Rbpj* loci making very unlikely to obtain pups that inherit a copy of the transgene without bearing at the same time the floxed *Rbpj* allele due to genetic linkage among them.

In order to fulfil to the best of our possibilities the request of this reviewer, we provide in the revised version of the manuscript a detailed comparison between control mice with normal levels of *Rbpj* (*Pdgfrb*-*CreERT2*^{+/+}) and littermates with hemizygous deletion of *Rbpj* (*Rbpj*^{+iPC}) (Supplementary Fig. 2). Most importantly, it is clear that mice with a single functional copy of *Rbpj* do not show haemorrhages nor changes in vascular area or mural cell coverage at any of the time points analysed (Supplementary Fig. 2a-d). Vessel patterning, expression of mural cell markers and ensheathment of astrocytic endfeet remain unaffected as well (Supplementary Fig. 2c-e). Moreover, phosphorylation of SMADs (Supplementary Fig. 2f, g) and activation of integrin β 1 (Supplementary Fig. 2h), which are severely affected in *Rbpj* conditional knockouts, are undistinguishable from Cre-negative controls.

14. The increased Thbs1 signal in Figure 4l appears to be in vessels larger than capillaries.

Reply: We have replaced the figure with a new set of confocal images showing higher Thbs1 immunosignals in mural cells associated to the microvasculature (Fig. 5l in the revised version of the manuscript).

15. The phSMAD3 staining in Figure 4b, and the phSMAD1/5 staining in Figure 4d appear to be nuclei that are too numerous to be pericyte nuclei. Instead they look like endothelial nuclei.

Reply: As the reviewer points out, the increased phosphorylation of SMAD3 and SMAD1/5 (Fig. 5b, d in the revised version), correspond to higher activation of these transcriptional regulators in the endothelium as further analysed in single-optical sections (Fig. 5f, g) and stated in the text: "...increased SMAD3 phosphorylation was a common feature of both *Rbpj*^{iPC} ECs and pericytes, whereas SMAD1/5 phosphorylation was only detected in mutant endothelium...". We have slightly rephrased this sentence to put more emphasis on this very important point regarding the changes in pericytes-endothelial cells crosstalk.

16. The authors RNA-seq analysis uses a cutoff of $\log_2(0.5)$; this is a liberal cutoff that may identify differential gene expression changes that are not real (applies to Fig 2).

Reply: We agree with the reviewer in that a lower cut-off increases the risk of including false positive genes, on the other hand, a very stringent cut-off may fail to detect relevant genes whose expression changes in a milder, yet meaningful, way. Any given fold-change threshold is arbitrary and there is no definitive agreement on which cut-off of \log_2 allows identification of "real" gene expression changes. Along the manuscript there are several steps of further validation of the differentially expressed genes detected by the RNA-Seq analysis, either by qPCR interrogation of freshly sorted cells or direct assessment of protein levels by immunostaining. Moreover, the vast majority of genes relevant for the pathophysiological characterization of the phenotype is based on fold changes which are way above $\log_2(0.5)$, reducing the possibilities of having included false positives. Finally, this revised version contains ChIP-Seq analysis for RBPJ binding (see reply to reviewer #2), which shows that the fundamental genes driving pathogenic transformation of pericytes through TGF β signalling upregulation are indeed *bona fide* RBPJ-bound targets, further excluding spurious identification of differentially expressed genes.

17. Numerous differentially expressed genes and pathways are presented. The authors could do a better job presenting this information more clearly/concisely for ease of the reader to follow the story and the authors' working model of how disrupting signaling mechanisms underlie the resulting CCM-like vascular phenotype.

Reply: We are thankful for this criticism and have therefore profoundly reorganized the way data is presented along the manuscript. Moreover, new pieces of data generated during the revision reinforce our working model and facilitate a step-by-step explanation of the changes elicited in mural cells after *Rbpj* inactivation. We hope this revised version meets the expectations of the reviewer and makes it easy for readers to follow the story.

18. The authors should provide supplementary spreadsheets with a full list of differentially expressed genes.

Reply: We have included as supplementary information spreadsheets where a full list of differentially expressed genes at P7 and P10 is included.

Minor comments:

1. The citation for Hall et al., should be moved just after the statement that, “Pericytes and, in particular, their contractility has implicated in the regulation of cerebral blood flow”, rather than being at the end of the sentence.

Reply: We have followed this reviewer’s suggestion since it helps to better reflect the discrepancy between different authors regarding pericyte contractility.

2. The authors should confirm that the representative images between groups are at the same magnification in Figure 3e and f.

Reply: We have confirmed that the representative images shown in Fig. 3e and 3f of the original manuscript (Fig. 4e, f in the revised version) are indeed of the same magnification. The perceived differences in vessel size are again a consequence of the vascular enlargement induced by the loss of *Rbpj* in mural cells.

Reviewer #2 (Remarks to the Author):

The manuscript by Diéguez-Hurtado and colleagues utilizes state of the art inducible genetics in mice to study the role of RBPJ, a transcriptional repressor and central effector of Notch signalling, selectively in post-natal pericytes and vascular smooth muscle cells in the brain. The authors carefully confirm specificity of the *Pdgfrb-CreERT2* driver line in the brain, and study the effects on brain vasculature, pericyte coverage, morphology, gene expression, as well as the neuronal environment following inducible deletion of RBPJ at P1,2 and 3. By P10, the authors find dramatic vascular defects in many but not all brain areas, and incipient changes at P7. The authors undertake a highly comprehensive phenotypical characterization, provide beautiful illustrations and a wealth of informative controls. They further investigate transcriptome of the pericytes using the ribotag approach. The authors find that vascular lesions that carry similarities with CCM associate with increased endothelial cell proliferation and pericyte gene expression changes indicating strong upregulation of TGF β and BMP signalling, increased matrix production including matrix components involved in TGF β binding and activation. They further provide convincing evidence for endothelial increases in phospho SMAD1/5 and both pericyte and endothelial increases in phospho SMAD2/3.

Surprisingly genetic inactivation of Notch2 and 3 in pericytes, or general Notch inactivation in pericytes by *dnMaml* expression, as well as activation of Notch using NICD do not phenocopy the RBPJ deletion effects. Neither does pericyte depletion using genetic DTA expression in the same population.

The authors therefore conclude that RBPJ has an important repressor role in postnatal pericytes in the brain, unlikely connected to Notch activity.

Intriguingly, the present work provides evidence for a detrimental effect of pericyte dysregulation as potential driver of vascular lesions and compromising the neurovascular unit in postnatal brains. As such, the work is highly noteworthy, and together with the sophisticated and detailed analysis, the wealth of data, it will be of great interest to both the specialist and general readership.

Reply: We deeply appreciate this reviewer's words and are thankful for the positive evaluation of our work.

The question whether or not this is a model that will help us understand potential deleterious effects of pericyte dysregulation in adulthood remains however open. Surprisingly, deleting RBPJ in pericytes in the adult brain does not cause any phenotype. This in itself is a striking observation, as one would expect that silencing of the gene loci that drive the changes in the juvenile brain would need to continue in order to protect vascular homeostasis in the adult. So why does RBPJ deletion in the adult have no effect? Is that because RBPJ is no longer actively transcribed, and there is a pool of RBPJ protein that is sufficient to suppress the loci in the adult quiescent state? Or do we need to suspect further chromatin changes that silence the same loci even in the absence of RBPJ protein? Answering these questions will help to assess whether the reported phenotype merely reflects a derailed differentiation process of pericytes caused by an artificially induced derepression, or whether the underlying biology might in fact predict important functions that are at play in human disease processes. I realize that answering these questions in detail experimentally may entail many new and different approaches to study chromatin and would go beyond the scope of the present work. In the absence of new data, it would seem however appropriate to discuss the reasons for no overt phenotype in the adult in a little more detail.

Reply: We thank the reviewer for the thoughtful comments. This question is obviously not only relevant for *Rbpj* in mural cells, but also for a long list of genes that are essential during certain stages of development but not for homeostasis in the healthy adult. The underlying reasons are usually not clear. Physiological requirements or fundamental molecular factors – such as the accessibility of certain chromatin region or the expression of other transcription factors may have changed. Alternatively, it is feasible that cells in a fully developed, mature quiescent context are comparably stable until they get reactivated in response to tissue damage or other insults, as is the case for *Rbpj* in mural cells after stroke.

While it is obviously not feasible to perform a detailed analysis of all potentially relevant factors, such as chromatin organization, protein turnover and signalling, we have generated new data in order to support our findings and suggest potential mechanisms explaining differences between juvenile and adult mice.

In one hand, we have performed ChIP-Seq analysis of RBPJ binding in cultured primary brain pericytes. These results, integrated in Fig. 6 and Supplementary Fig. 12, show that RBPJ binding sites are enriched for either the promoter mark H3K4me3 or the enhancer mark H3K4me1, indicating association of RBPJ to genomic areas related to transcriptional regulation. Among the RBPJ-bound genes identified, we found 122 genes that are also upregulated *in vivo* in *Rbpj*-mutant mice. This list includes critical regulators of TGFβ signalling and neuroinflammation such as *Thbs1*, *Tgfb3* and *Ccl2*, further indicating that RBPJ is directly involved in the regulation of these transcriptional domains. Unfortunately, the conditions associated to *in vitro* culture of pericytes (confluency, monoculture, lack of 3D organization, etc.) preclude a more detailed analysis of histone marks that may reflect repressive or active status of chromatin or to assess age-specific differences.

On the other hand, we have done a careful analysis of Notch signalling activation by using two different reporter mouse lines: *Hey1-EGFP* and *Rbpj-H2B-Venus*, which reflect gene transcription downstream of Notch either by revealing expression of the Notch-target *Hey1* or by driving expression of nuclear-localized Venus under control of consensus RBPJ responsive elements coupled to a basal promoter. Interestingly, we could not find evidence of Notch signalling activation (i.e., no *Hey1-EGFP* or nuclear Venus expression) in brain pericytes of young pups from P1 to P15 (Supplementary Fig. 7a, b), further reinforcing the view that RBPJ mediates transcriptional repression in postnatal pericytes. Noteworthy, the landscape of Notch signalling activation in mature animals is strikingly different, with frequent detection of nuclear Venus expression in *bona fide* pericytes of the brain microvasculature (Revision Fig. 5) suggesting that, once the vasculature has matured, RBPJ is more frequently engaged in canonical Notch signalling. Moreover, we have not detected any strong changes in *Rbpj* expression in 7-week old mice when compared to young pups using *Pdgfrb-CreERT2*-mediated RiboTag-labelling (data not shown), indicating that drastic reduction in *Rbpj* transcription is unlikely to play an important role. As now stated in the revised manuscript, we propose that *Rbpj*-mediated repression is critical during specific phases of postnatal development or regenerative processes associated to blood vessels remodelling and fate-determination in mural cells. As such, in immature or activated pericytes, RBPJ would prevent expression of loci which may promote premature specification or final commitment to a given cellular identity (i.e. vascular smooth muscle cell) and would therefore allow cellular plasticity in order to accommodate the dynamic behaviour of vessel remodelling. In the quiescent vasculature, where the maturation process

Revision Figure 5. Analysis of canonical Notch signalling activation in brain mural cells from mature animals.

a, Confocal images showing frequent detection of nuclear Venus fluorescence (green) in $PDGFR\beta^+$ (white) pericytes in the brain cortex of 7-month old mice. Blood vessels are labelled with IB4 (red). Scale bar, 50 μ m. Bottom panel shows single optical sections of individual pericytes for better appreciation of the nuclear fluorescence. Numbers correspond to the highlighted pericytes which are positive for the reporter (yellow arrowheads). Pericytes that do not show reporter activation are marked with white arrowheads.

is complete and proper cellular identity reaches final differentiation, the specific RBPJ-mediated repression is no longer necessary either because expression of the repressed loci is necessary for commitment or because other, yet uncharacterized, regulatory mechanisms take over this role. Instead, in such committed mural cells, canonical Notch signalling appears relevant for maintenance of the differentiated phenotype, as has been proposed previously²⁸.

The final observation of increased stroke lesions is less surprising, but does indicate that an activation state is a prerequisite for the strong effects of pericytic RBPJ deletion. Here may lie the opportunity to at least comment on whether increased RBPJ protein turnover/expression in pericytes is part of the activation state and may therefore provide a technical explanation for the lack of phenotype in adult.

Reply: We agree with this reviewer's interpretation in the sense that an activation state (angiogenesis and vessel remodelling after ischemic stroke) in pericytes is a setting where RBPJ-mediated gene repression plays fundamental roles. This may reflect increased cellular plasticity in pericytes after stroke and therefore a less committed phenotype, which would resemble the higher cellular plasticity of mural cells during early postnatal development. Recent RNA-Seq data made available in Gene Expression Omnibus (GSE114652), where the brain of mice subjected to MCAO-induced stroke vs. sham-operated animals was analysed by RNA-Seq, did not reveal significant changes in the expression level of *Rbpj* (in total RNA). This result suggests that the relevance of RBPJ in mural cells upon ischemic insult is probably not mediated by changes in its expression.

Minor comments and corrections:

Line 66: word missing "has been implicated"

Reply: We thank this reviewer for spotting this. The text is now corrected.

Line 162: the authors probably mean "sensitive" not "sensible"

Reply: The text is now corrected.

Line 206, figure 2 g,h Does this indicate partial differentiation of pericytes towards VSM cells? Is it a consequence of detachment? does it mean that VSM usually are lower in RBPj expression or is this because Notch is active in VSM and therefore these loci not repressed?

Reply: Indeed, molecular profiling of *Rbpj* mutant mural cells suggests the acquisition of an arterial SMC-like phenotype given the consistent upregulation of genes which are prominently expressed in these cells with respect to pericytes, according to recently generated single cell-RNA-Seq data¹. Moreover, that study has proposed that pericytes and venous SMCs are profoundly different from SMCs associated with arteries and arterioles. Since the phenotypic changes in pericytes vs. arterial SMCs upon *Rbpj* deletion are somehow antagonistic, it is tempting to propose a different regulatory role for RBPJ in different mural cell populations. Interestingly, *Rbpj* expression levels are significantly higher (>1.7-fold) in arterial SMCs with respect to mid-capillary pericytes in the brain of rats²⁹. Taking advantage of the *Rbpj-H2B-Venus* mouse line, we have analysed the activation of RBPJ-mediated transcription as a readout of canonical Notch signalling in brain arteries of P10 mice. Strikingly, both penetrating cortical arterioles as well as pial arteries in the surface of the brain showed consistent and high expression of nuclear Venus in their SMCs (Supplementary Fig. 7c, d), in clear contrast to the absence of reporter signals in pericytes of the microvasculature (Supplementary Fig. 7a, b). These results indicate that RBPJ during early postnatal development plays different roles in pericytes and arterial SMCs, acting prominently as a repressor in the former and mediating Notch-driven transcription in the latter, where it has well documented roles in promoting differentiation and contractility³⁰.

Line 255, word missing after "post-natal"

Reply: Thanks for the observation. The text is now corrected.

Line 261, should read “normally”

Reply: Thanks for the observation. The text is now corrected.

Line 328, word missing

Reply: Thanks for the observation. The text is now corrected.

Line 330, spelling metalloprotease

Reply: Thanks for the observation. The text is now corrected.

Line 381, What happens to endothelial cell proliferation after stroke in the mutants?

Reply: We could not detect obvious differences in the overall proliferation rate after stroke in mutants vs. control mice as shown in the Revision Fig. 6. Specific interrogation of endothelial cell proliferation is precluded by the limited availability of compatible antibodies against Ki67 and the EC-nuclei marker Erg1 from different species. We regret not being able to give a more conclusive answer in this regard.

Line 398, are there quantitative differences in RBPj expression levels between pericytes and vascular smooth muscle on arteries, veins and capillaries that could explain the differential effects on the different vessel segments?

Reply: This question has been addressed in a previous answer and evidences implying important differences in the activation status of Notch signalling between pericytes and arterial SMCs are now included as part of the revised manuscript (Supplementary Fig. 7).

Revision Figure 5. Analysis of EC proliferation in brain blood vessels from control and *Rbpj*^{IP} mice after stroke.

a. Confocal images showing no obvious changes in the total amount of proliferative (Ki67⁺, white) cells in the vasculature (ICAM2, red) of control or *Rbpj*-mutant animals. Recombinant mural cells are detected in virtue of GFP expression (green). Scale bar, 100 μ m.

Line 406, this discussion again raises the question whether RBPj is naturally low in arterial vSMC, possibly explaining the lack of phenotype in arteries in the study.

Reply: As already mentioned, *Rbpj* expression is higher in arterial SMCs relative to pericytes, at least in the rat brain²⁹. It should be noted that absence of haemorrhagic lesions around arteries does not imply a lack of phenotype in mural cells around these vessels. Indeed, we report that SMC coverage around arteries is diminished and irregular, consistent with the

previously proposed role of Notch signalling in these cells³⁰. As previously highlighted, we propose that differences between pericytes and arterial SMCs with respect to *Rbpj* deletion are based on the distinct levels of Notch activation among them.

Reviewer #3 (Remarks to the Author):

This is a comprehensive and rigorous study concerning the role of this transcription factor in pericyte biology. Combining the main data and the extended data provided, this manuscript required considerable time and labor to read through the wealth of figures and data. This is not a criticism, merely a statement of the comprehensive nature of the study.

As stated above, I found the work to be quite comprehensive and rigorous. The rationale for each experiment was very clear and the data was clearly described and illustrated (although toggling back and forth between the main data and the extended data was quite confusing). I have no scientific concerns regarding the work.

Reply: We sincerely appreciate the assessment of our work by the reviewer and agree with his observations. In order to ease going through the manuscript we have done substantial changes in the organization of the figures and main text.

My only concern is that if these vascular lesions truly resemble cavernous malformations, it is surprising that no one has reported this gene to be mutated in any families or individual cases.

Reply: In order to address the existence of *Rbpj* mutations in CCM patients, surgically resected samples from multiple sporadic lesions in 16 individuals, for which there was no known relative affected and no detected mutation in *Krit1*, *Ccm2* or *Pdcd10*, were sequenced by Sanger in collaboration with Dr. Florence Riant and Dr. Elisabeth Tournier-Lasserre (Paris, France). Although this analysis was not able to detect mutations in *Rbpj*, suggesting that this is not a frequent trigger of CCM disease in humans, we should point out that any expected mutation in *Rbpj* should be somatic since germ-line mutations induce the Adams-Oliver Syndrome, a multiple-malformation disorder³¹, or are embryonic lethal. Given the difficulties in the identification of low-level mosaicism by conventional Sanger sequencing, it is not possible to completely rule out potential involvement of *Rbpj*-deficient mural cells in the pathogenesis of CCMs.

Furthermore, the Discussion section does not discuss how this work relates to CCM, other than that the lesions resemble CCMs. How does this inform us of CCM pathobiology and what does it mean that a TF in pericytes can cause the same vascular lesions (with some of the same molecular signatures) as loss of CCM proteins in the endothelial cell compartment. If the case is to be made that these lesions resemble CCMs, then it would be important to discuss this discovery in the context of the rich literature on CCM pathobiology.

Reply: We agree with this reviewer that it is necessary to discuss particular aspects of CCM pathobiology in which defective pericytes may have an impact. We have therefore edited the Discussion section along the following lines. CCM proteins interact with several molecules which have been shown to drive different aspects of pathogenic transformation in ECs.

Interestingly, the most relevant signalling pathways proposed to induce lesion formation, namely increased RhoA GTPase activity³², augmented MEKK3/KLF2,4 signalling^{33,34}, abnormal activation of the TGF β /BMP/SMAD pathway²² or aberrant activation of integrin β 1 upstream of KLF2 overexpression³⁵, may be deregulated as a consequence of molecular mechanisms that do not involve CCMs. As such, several different stimuli are able to drive the activation of these signalling pathways and we show pericytes may have an important role in controlling them. The morphological and molecular resemblance of CCMs by mural cell-specific *Rbpj* deletion may not fully recapitulate the nature of this disease, but clearly reflect that deleterious pericytes are able to trigger aberrant signalling programs which compromise EC behaviour and jeopardize vascular integrity.

References

- 1 Vanlandewijck, M. *et al.* A molecular atlas of cell types and zonation in the brain vasculature. *Nature* **554**, 475-480, doi:10.1038/nature25739 (2018).
- 2 Winkler, E. A., Bell, R. D. & Zlokovic, B. V. Pericyte-specific expression of PDGF beta receptor in mouse models with normal and deficient PDGF beta receptor signaling. *Mol Neurodegener* **5**, 32, doi:10.1186/1750-1326-5-32 (2010).
- 3 Saunders, A. *et al.* Molecular Diversity and Specializations among the Cells of the Adult Mouse Brain. *Cell* **174**, 1015-1030 e1016, doi:10.1016/j.cell.2018.07.028 (2018).
- 4 Cuervo, H. *et al.* PDGFRbeta-P2A-CreER(T2) mice: a genetic tool to target pericytes in angiogenesis. *Angiogenesis* **20**, 655-662, doi:10.1007/s10456-017-9570-9 (2017).
- 5 Imayoshi, I., Sakamoto, M., Yamaguchi, M., Mori, K. & Kageyama, R. Essential roles of Notch signaling in maintenance of neural stem cells in developing and adult brains. *J Neurosci* **30**, 3489-3498, doi:10.1523/JNEUROSCI.4987-09.2010 (2010).
- 6 Sato, C. *et al.* Loss of RBPj in postnatal excitatory neurons does not cause neurodegeneration or memory impairments in aged mice. *PLoS One* **7**, e48180, doi:10.1371/journal.pone.0048180 (2012).
- 7 Ganat, Y. M. *et al.* Early postnatal astroglial cells produce multilineage precursors and neural stem cells in vivo. *J Neurosci* **26**, 8609-8621, doi:10.1523/JNEUROSCI.2532-06.2006 (2006).
- 8 Clatterbuck, R. E., Eberhart, C. G., Crain, B. J. & Rigamonti, D. Ultrastructural and immunocytochemical evidence that an incompetent blood-brain barrier is related to the pathophysiology of cavernous malformations. *J Neurol Neurosurg Psychiatry* **71**, 188-192 (2001).
- 9 Wong, J. H., Awad, I. A. & Kim, J. H. Ultrastructural pathological features of cerebrovascular malformations: a preliminary report. *Neurosurgery* **46**, 1454-1459 (2000).
- 10 McDonald, D. A. *et al.* A novel mouse model of cerebral cavernous malformations based on the two-hit mutation hypothesis recapitulates the human disease. *Hum Mol Genet* **20**, 211-222, doi:10.1093/hmg/ddq433 (2011).
- 11 Gerhardt, H., Wolburg, H. & Redies, C. N-cadherin mediates pericytic-endothelial interaction during brain angiogenesis in the chicken. *Dev Dyn* **218**, 472-479, doi:10.1002/1097-0177(200007)218:3<472::AID-DVDY1008>3.0.CO;2-# (2000).
- 12 Paik, J. H. *et al.* Sphingosine 1-phosphate receptor regulation of N-cadherin mediates vascular stabilization. *Genes Dev* **18**, 2392-2403, doi:10.1101/gad.1227804 (2004).
- 13 Li, F. *et al.* Endothelial Smad4 maintains cerebrovascular integrity by activating N-cadherin through cooperation with Notch. *Dev Cell* **20**, 291-302, doi:10.1016/j.devcel.2011.01.011 (2011).

- 14 Tanriover, G. *et al.* Ultrastructural analysis of vascular features in cerebral cavernous malformations. *Clin Neurol Neurosurg* **115**, 438-444, doi:10.1016/j.clineuro.2012.06.023 (2013).
- 15 Lackovic, V. *et al.* Ultrastructural Analysis of Small Blood Vessels in Skin Biopsies in Cadasil. *Arch Biol Sci* **60**, 573-580, doi:10.2298/Abs08045731 (2008).
- 16 Claudio, L. Ultrastructural features of the blood-brain barrier in biopsy tissue from Alzheimer's disease patients. *Acta Neuropathol* **91**, 6-14 (1996).
- 17 Andrae, J., Gallini, R. & Betsholtz, C. Role of platelet-derived growth factors in physiology and medicine. *Genes Dev* **22**, 1276-1312, doi:10.1101/gad.1653708 (2008).
- 18 Funa, K. & Sasahara, M. The roles of PDGF in development and during neurogenesis in the normal and diseased nervous system. *J Neuroimmune Pharmacol* **9**, 168-181, doi:10.1007/s11481-013-9479-z (2014).
- 19 Batra, S., Lin, D., Recinos, P. F., Zhang, J. & Rigamonti, D. Cavernous malformations: natural history, diagnosis and treatment. *Nat Rev Neurol* **5**, 659-670, doi:10.1038/nrneuro.2009.177 (2009).
- 20 Spiegler, S., Rath, M., Paperlein, C. & Felbor, U. Cerebral Cavernous Malformations: An Update on Prevalence, Molecular Genetic Analyses, and Genetic Counselling. *Mol Syndromol* **9**, 60-69, doi:10.1159/000486292 (2018).
- 21 McDonald, D. A. *et al.* Lesions from patients with sporadic cerebral cavernous malformations harbor somatic mutations in the CCM genes: evidence for a common biochemical pathway for CCM pathogenesis. *Hum Mol Genet* **23**, 4357-4370, doi:10.1093/hmg/ddu153 (2014).
- 22 Maddaluno, L. *et al.* EndMT contributes to the onset and progression of cerebral cavernous malformations. *Nature* **498**, 492-496, doi:10.1038/nature12207 (2013).
- 23 Benedito, R. *et al.* Notch-dependent VEGFR3 upregulation allows angiogenesis without VEGF-VEGFR2 signalling. *Nature* **484**, 110-114, doi:10.1038/nature10908 (2012).
- 24 Benedito, R. *et al.* The notch ligands Dll4 and Jagged1 have opposing effects on angiogenesis. *Cell* **137**, 1124-1135, doi:10.1016/j.cell.2009.03.025 (2009).
- 25 Augustin, H. G., Koh, G. Y., Thurston, G. & Alitalo, K. Control of vascular morphogenesis and homeostasis through the angiopoietin-Tie system. *Nat Rev Mol Cell Biol* **10**, 165-177, doi:10.1038/nrm2639 (2009).
- 26 Hall, C. N. *et al.* Capillary pericytes regulate cerebral blood flow in health and disease. *Nature* **508**, 55-60, doi:10.1038/nature13165 (2014).
- 27 Gould, I. G., Tsai, P., Kleinfeld, D. & Linninger, A. The capillary bed offers the largest hemodynamic resistance to the cortical blood supply. *J Cerebr Blood F Met* **37**, 52-68, doi:10.1177/0271678x16671146 (2017).
- 28 Liu, H., Kennard, S. & Lilly, B. NOTCH3 expression is induced in mural cells through an autoregulatory loop that requires endothelial-expressed JAGGED1. *Circ Res* **104**, 466-475, doi:10.1161/CIRCRESAHA.108.184846 (2009).
- 29 Chasseigneaux, S. *et al.* Isolation and differential transcriptome of vascular smooth muscle cells and mid-capillary pericytes from the rat brain. *Sci Rep* **8**, 12272, doi:10.1038/s41598-018-30739-5 (2018).
- 30 Baeten, J. T. & Lilly, B. Notch Signaling in Vascular Smooth Muscle Cells. *Adv Pharmacol* **78**, 351-382, doi:10.1016/bs.apha.2016.07.002 (2017).
- 31 Hased, S. J. *et al.* RBPJ mutations identified in two families affected by Adams-Oliver syndrome. *Am J Hum Genet* **91**, 391-395, doi:10.1016/j.ajhg.2012.07.005 (2012).
- 32 Stockton, R. A., Shenkar, R., Awad, I. A. & Ginsberg, M. H. Cerebral cavernous malformations proteins inhibit Rho kinase to stabilize vascular integrity. *J Exp Med* **207**, 881-896, doi:10.1084/jem.20091258 (2010).

- 33 Cuttano, R. *et al.* KLF4 is a key determinant in the development and progression of cerebral cavernous malformations. *EMBO Mol Med* **8**, 6-24, doi:10.15252/emmm.201505433 (2016).
- 34 Zhou, Z. *et al.* Cerebral cavernous malformations arise from endothelial gain of MEKK3-KLF2/4 signalling. *Nature* **532**, 122-126, doi:10.1038/nature17178 (2016).
- 35 Renz, M. *et al.* Regulation of beta1 integrin-Klf2-mediated angiogenesis by CCM proteins. *Dev Cell* **32**, 181-190, doi:10.1016/j.devcel.2014.12.016 (2015).

REVIEWERS' COMMENTS:

Reviewer #1 (Remarks to the Author):

The authors' new data and analyses have substantially strengthened the manuscript. Specifically, they added a new control group to support that no phenotype is observed in the single allele deletion of Rbpj(Rbpj^{+/iPC}). They also greater characterize the observed vascular phenotype by adding new analysis of EC-mural cell crosstalk, TJ and AJ proteins, and new in vivo multiphoton microscopy analysis of blood flow velocity along the arteriovenous axis. New supplementaal spreadsheets of differentially expressed genes are now included. Overall, the study is greatly improved and I have no further suggestions.

Reviewer #2 (Remarks to the Author):

The authors have further improved this impressive study and addressed all my concerns. As far as I can tell they have more than addressed also all the detailed questions raised by the other reviewers. This is a very comprehensive analysis with a wealth of data, and substantial conceptual advance that will not only be of interest to a wide readership, but also likely stimulate further studies to understand the mechanisms of pericyte and vSMC maturation and quiescence, and their adaptive regulation post injury.

I recommend publication with no delay.

Sincerely,

Holger Gerhardt

Reviewer #3 (Remarks to the Author):

The authors have addressed my concerns.

POINT-BY-POINT RESPONSE TO REVIEWER'S COMMENTS

REVIEWERS' COMMENTS:

Reviewer #1 (Remarks to the Author):

The authors' new data and analyses have substantially strengthened the manuscript. Specifically, they added a new control group to support that no phenotype is observed in the single allele deletion of Rbpj(Rbpj+/iPC). They also greater characterize the observed vascular phenotype by adding new analysis of EC-mural cell crosstalk, TJ and AJ proteins, and new in vivo multiphoton microscopy analysis of blood flow velocity along the arteriovenous axis. New supplementaal spreadsheets of differentially expressed genes are now included. Overall, the study is greatly improved and I have no further suggestions.

Reply: We appreciate the positive evaluation of our work and thank the reviewer for his valuable suggestions, which have helped us to improve our manuscript.

Reviewer #2 (Remarks to the Author):

The authors have further improved this impressive study and addressed all my concerns. As far as I can tell they have more than addressed also all the detailed questions raised by the other reviewers. This is a very comprehensive analysis with a wealth of data, and substantial conceptual advance that will not only be of interest to a wide readership, but also likely stimulate further studies to understand the mechanisms of pericyte and vSMC maturation and quiescence, and their adaptive regulation post injury. I recommend publication with no delay.

Sincerely,

Holger Gerhardt

Reply: We are very grateful for the enthusiastic assessment of our study and the reviewer's kind words regarding the impact of our work in the field.

Reviewer #3 (Remarks to the Author):

The authors have addressed my concerns.

Reply: We would like to take the opportunity to thank the reviewer for his thoughtful comments and suggestions.